# ALKBH1 activity *in vitro* and human cell lines by isotope dilution mass spectrometry

Bennett Henzeler[1], Olga Hofmeister[1], Ken Kögel[1], Yuyang Qi[3], Felix Hagelskamp[1], Matthias Heiß[1], Florian Schelter[1], Felix Xu[1], Lena J. Daumann[2], Thomas Carell[1], Stefanie Kaiser[3]*, Sabine Schneider[1]*

1 Department of Chemistry, Institute for Chemical Epigenetics, Ludwig-Maximilians Universität Munich, Munich, Germany, 2 Chair for Bioinorganic Chemistry, Heinrich-Heine-Universität Düsseldorf, Düsseldorf, Germany, 3 Institute for Pharmaceutical Chemistry and Pharmaceutical Analytic, Goethe Universität Frankfurt, Frankfurt, Germany

☉ These authors contributed equally.
* stefanie.kaiser@pharmchem.uni-frankfurt.de (SK); sabine.schneider@cup.lmu.de (SS)

## Abstract

AlkB homolog 1 (ALKBH1) is a member of the AlkB family of Fe(II) and α-ketoglutarate (α-KG)-dependent dioxygenases, known for its enzymatic activity on various nucleic acid substrates. Its reported targets include N1-methyladenosine (m$^1$A), N6-methyladenosine (m$^6$A), N3-methylcytidine (m$^3$C), and 5-methylcytosine (m$^5$C) in RNA, as well as N6-methyladenine (N$^6$-mA, 6mA) in DNA and the histone protein H2A. Moreover, dysregulation or dysfunction of ALKBH1 has been implicated in a broad spectrum of human diseases. In order to shed further light on the substrate scope and role of ALKBH1, we used quantitative mass spectrometry to assess its activity *in vitro* and in human cell lines. To study ALKBH1 activity on defined substrates *in vitro*, we enzymatically generated tRNAs specifically carrying the m$^3$C32 (tRNA-Thr$^{UGU}$ and tRNA-Ser$^{UCN}$) and i$^6$A37 (tRNA-Ser$^{UCN}$) modifications. Here we show that ALKBH1 reduces m$^3$C, m$^1$A and m$^5$C *in vitro* in total extracts of tRNAs, but has no impact on rRNA modifications in human cell lines. However, upon overexpression or siRNA-mediated knock-down of ALKBH1 in human cell lines no impact on the modification of total tRNA extracts or specifically enriched RNAs could be observed. In addition, varying the glucose and fetal bovine serum (FBS) concentration in the growth medium of HEK293T cells, in combination with ALKBH1 siRNA-mediated knock-down, also shows no impact on tRNA methylation. Based on our data, we conclude that in human cells lines grown under optimal conditions ALKBH1 does not play an important role in the demethylation of tRNAs.

## Introduction

RNA modifications play a crucial role in shaping the three-dimensional structure, stability, and interaction networks of RNA molecules, thereby acting as important

**Data availability statement:** All relevant data are within the paper and its Supporting information files.

**Funding:** This study was supported by the German Research Foundation (DFG) in the form of a grant awarded to S.S., S. K., L.D. and T.C. (SFB 1309; project number 325871075) and a grant awarded to S.S. (SCHN 1273-9) in the form of a salary for O.H., K.K., Y.Q, F.H and F.S.. Moreover this study was supported by the Federal Ministry of Research, Technology and Space (BMFTR) in the form of a grant awarded to S.S. (Cluster for Nucleic Acid Therapeutics Munich, CNATM, ID: 03ZU1201AA) in the form of a salary for B.H.. The specific roles of this author are articulated in the 'author contributions' section. The funders had no role in study design, data collection and analysis, decision to publish, or preparation of the manuscript.

**Competing interests:** No authors have competing interests.

regulators of gene transcription and translation and ultimately influencing gene expression [1]. Transfer RNAs (tRNAs) and ribosomal RNAs (rRNAs) represent two RNA species that are particularly heavily modified, which is essential for their characteristic structures and functions. In tRNAs, modifications at the wobble position are central to maintaining translation fidelity and efficiency [2,3]. Additionally, modifications within the anticodon arm further contribute to tRNA stability and structural integrity, indirectly regulating protein biosynthesis through the modulation of translation efficiency [4]. Hence, dynamic regulation of tRNA modifications exerts a profound impact on translational efficiency and, consequently, protein expression. These modifications are catalyzed by RNA-modifying enzymes, commonly referred to as "writers" and "erasers". An important class of erasers includes members of the AlkB family of Fe(II)/α-ketoglutarate (α-KG)-dependent dioxygenases [5,6]. Among them, ALKBH1 has been reported to catalyze the removal of various modifications, including N6-methyladenosine ($m^6A$) [7], N1-methyladenosine ($m^1A$) [8,9], N3-methylcytidine ($m^3C$) [10], 5-methylcytosine ($m^5C$) [11,12], N6-methyladenine ($N^6$-mA, 6mA) [13], as well as modifications on histone H2A [14]. The catalytic versatility of ALKBH1 has been linked to diverse human disorders and proposed to act as a driver for oncogensis [7,9,11,14–16]. However, its precise enzymatic activities and biological functions remain a subject of ongoing debate. For example, the claim that ALKBH1 demethylates 6mA in DNA has been challenged [17,18]. We have previously demonstrated that ALKBH1 localizes to mitochondrial RNA granules, and that knock-down of ALKBH1 increases oxygen consumption in human cells. This phenotype is associated with altered oxidative phosphorylation (OXPHOS) and induction of the mitochondrial unfolded protein response (UPR) [19]. In the present study we investigate the activity of ALKBH1 *in vitro* and in human cell lines using quantitative isotope-labelled dilution mass spectrometry (quantitative LC-MS/MS) [20,21].

Recombinant ALKBH1 was expressed and purified to assess its activity on total tRNA and rRNA isolated from human cell lines. In these assays, ALKBH1 reduced the levels of $m^1A$, $m^3C$ and $m^5C$ in tRNAs, while no detectable effect was observed on N6-methyladenosine ($m^6A$). In eukaryotes, $m^3C$ is present at position 32 ($m^3C32$) in the anticodon arm of mitochondrial tRNA-Thr[UGU] and tRNA-Ser[UCN] (mt-RNAs) species, where it regulates protein expression by stabilizing the tRNA structure and modulating the kinetics of protein biosynthesis. Recently, an isoform of the RNA methyltransferase METTL8 was shown to localize to mitochondria in human cells, where it catalyzes methylation at C32 in mitochondrial tRNA-Ser[UCN] and tRNA-Thr[UGU]. This modification contributes to the coordinated expression of mitochondrially encoded proteins, supporting the efficient function of the respiratory chain [22,23]. Since ALKBH1 did not exhibit activity on a synthetic anticodon-arm mimic containing $m^3C$, we established an *in vitro* system to generate tRNAs with defined modifications. Specifically, *in vitro*-transcribed tRNA was enzymatically modified by METTL8 to introduce $m^3C32$ and by *Escherichia coli* MiaA to introduce N6-isopentenyladenosine ($i^6A$) at position 37. Using this substrate, ALKBH1 was able to reduce the $m^3C$ signal by approximately 20% *in vitro*. However, when we analyzed endogenous mitochondrial tRNAs isolated from HEK293T cells either overexpressing ALKBH1 or subjected

to siRNA-mediated ALKBH1 knock-down, no significant differences in the modification spectra were detected by quantitative LC-MS/MS. Previous studies have reported that ALKBH1 expression is influenced by glucose availability, suggesting a metabolic link to ALKBH1-mediated regulation of nucleobase modifications, particularly m$^1$A [8]. To address this, we cultured HEK293T cells under varying glucose concentrations and serum conditions and quantified RNA modifications by LC-MS/MS. Consistently, no significant alterations in tRNA or rRNA modification profiles were detected. Collectively, these results demonstrate that defined tRNA substrates carrying site-specific modifications can be enzymatically generated, providing a valuable platform for dissecting the activity of RNA modification "eraser" enzymes. Although ALKBH1 demonstrates demethylase activity toward multiple modified nucleobases *in vitro*, our findings suggest that it does not significantly influence the tRNA modification profile in HEK293T cells under homeostatic growth conditions

## Results

### ALKBH1 demethylates total tRNA but not rRNA *in vitro*

To gain insight into the substrate scope of ALKBH1, we heterologously expressed and purified ALKBH1 in *E. coli* (S1 Fig). We next incubated ALKBH1 with total tRNA and rRNA extracted from HEK293T cells in the presence of α-KG and Fe(II). Following treatment, the RNAs were digested to single nucleoside level and changes in the nucleoside modification spectra were quantified using quantitative LC-MS/MS [24]. This showed that ALKBH1 reduces m$^1$A, m$^5$C and m$^3$C levels in tRNA but does not affect rRNA modifications (Fig 1.)

After ensuring the successful expression of active ALKBH1, we wanted to further evaluate the ALKBH1 activity *in vitro*. For this, we used the commercial Succinate-Glo™ Assay (Promega), which indirectly measures Fe(II)/α-KG dependent dioxygenase activity by converting the generated succinate into ATP, which serves as a substrate for luciferase [25,26]. For this assay, auto-hydroxylation [27] of ALKBH1 may bias the results and thus we first designed ALKBH1 mutant constructs which are either catalytically inactive (AxA, catalytic His and Asp residues exchanged for Ala) or auto-hydroxylation-deficient (I218A) as controls. The use of these control constructs on total tRNA confirmed that succinate generation is not only correlated with catalytic activity but also- with auto-hydroxylation (S2 Fig).

Since ALKBH1 has also been shown to localize to the mitochondria [19,28], we utilized not only total tRNA, but also a synthetic RNA oligonucleotide mimicking the anticodon arm of the mt-tRNA-Ser$^{UCA}$, which carries the m$^3$C modification at position 32 [23,29]. This synthetic tRNA-anticodon arm mimic does not appear to be a suitable substrate for ALKBH1, as no changes in succinate levels were observed upon addition of ALKBH1$^{I218}$, which cannot undergo auto-hydroxylation (S2 Fig). We estimate that ALKBH1 requires certain structural elements of full-length tRNA, which could not form in the short, anticodon-mimicking fragments.

### ALKBH1 demethylates m$^3$C32 in *in vitro* transcribed tRNA

Following our previous observation that length and potentially structure are discriminators for ALKBH1 activity, we used *in vitro* transcription to generate longer, but defined substrates for testing ALKBH1 activity (S3 Fig). We initially introduced random methylation by treating the *in vitro* transcribed tRNA and human total tRNA with methyl methanesulfonate (MMS), which introduces damage-derived m$^1$A, m$^7$G and m$^3$C at random positions [30]. Afterwards, we incubated the damaged tRNAs with ALKBH1 and quantified the abundance of modified nucleoside by quantitative LC-MS/MS. Here we can observe a slight reduction in the abundance of m$^1$A, m$^3$C and m$^7$G in the MMS treated ct-tRNA-Val$^{AAC}$ (S4 Fig) which indicates that ALKBH1 recognizes in case of RNA damage the chemical structure of its substrate rather than the position within the tRNA. To shed light on ALKBH1's activity towards native m$^3$C, we obtained a defined tRNA modified at position 32, by expressing and purifying the methyltransferase METTL8. METTL8 was previously shown to methylate C32 at position N3 in mitochondrial tRNAs, specifically mt-tRNA-Ser$^{UCA}$ and mt-tRNA-Thr$^{UGU}$ [22]. However, since methylation of mt-tRNA-Ser$^{UCA}$ by METTL8 depends on the i$^6$A modification at A37 [23], we also expressed and purified MiaA, the tRNA

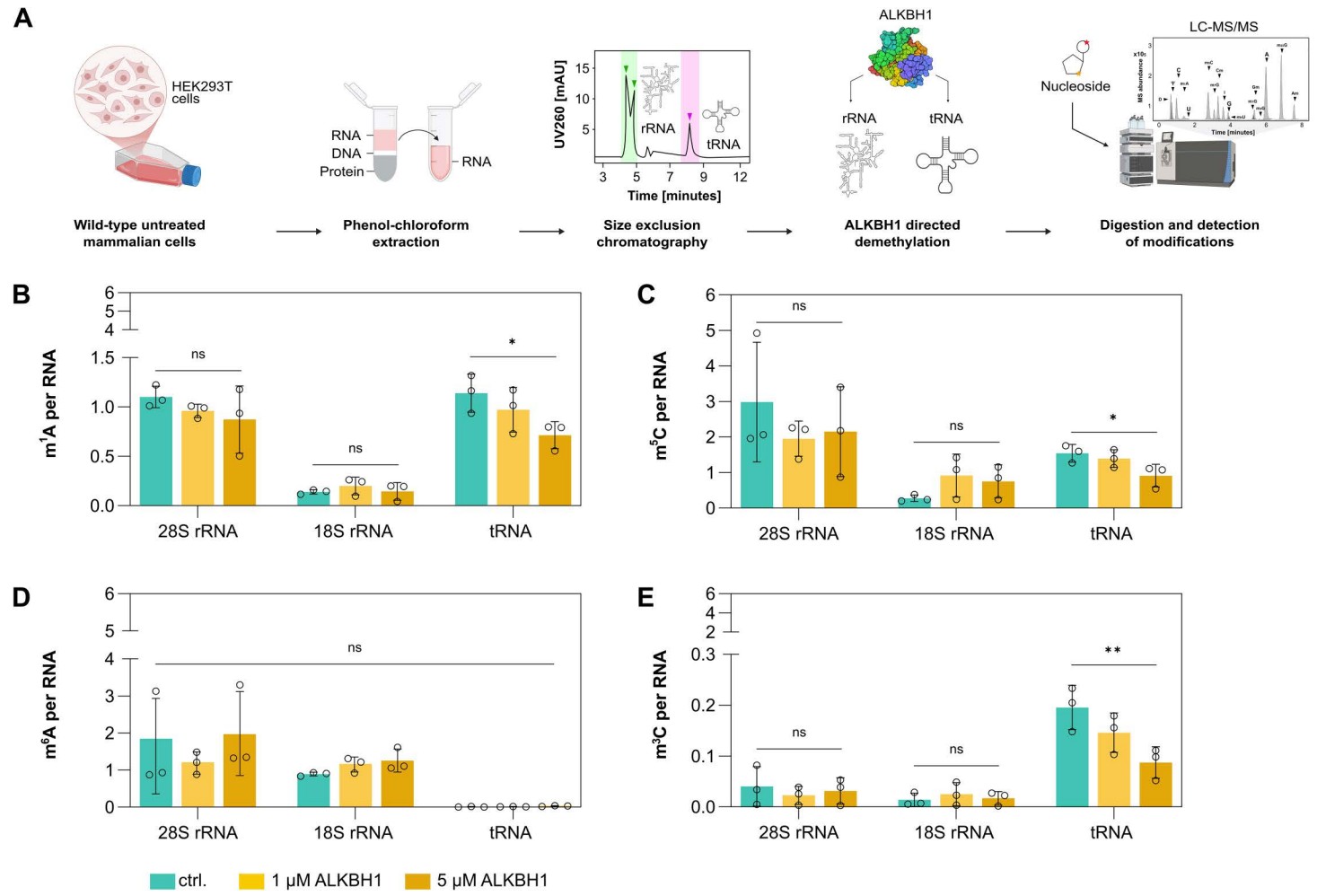

**Fig 1. Substrate specific demethylation activity of ALKBH1. A.** Schematic representation of RNA size-based separation and demethylation by ALKBH1 heterologous expressed in *E. coli*. Total RNA is extracted from HEK293T cells and separated by size into distinct fractions corresponding to 28S rRNA, 18S rRNA and tRNA. These RNA species are then subjected to *in vitro* demethylation using ALKBH1 heterologous expressed in *E. coli* under defined reaction conditions. Following enzymatic treatment, the RNA is processed and analyzed for the detection of RNA modifications using LC-MS/MS. B-E. Absolute m$^1$A **(B)**, m$^5$C **(C)**, m$^6$A (D) and m$^3$C (E) content in 28S rRNA, 18S rRNA and tRNA post *in vitro* treatment with ALKBH1 heterologous expressed in *E. coli* at concentrations of 0 µM, 1 µM and 5 µM. Error bars in the graph represent the $\pm$ mean **S.**E. (standard error) of three biological replicates. p-value ($p > 0.05 = $ ns, $p \leq 0.05 = *$, $p \leq 0.01 = **$, $p \leq 0.001 = ***$ and $p \leq 0.0001 = ****$) mentioned in the text is calculated by one-way ANOVA and indicates significant differences in means (Created in BioRender. Henzeler, **B.** (2025)).

dimethylallyltransferase from *E. coli*. (S1 Fig). Thus, we introduced the i$^6$A modification into mt-tRNA-Ser$^{UCA}$ using EcMiaA, followed by methylation of N3 at C32 by METTL8. METTL8 directly introduces the m$^3$C at C32 in mt-tRNA-Thr$^{UGU}$. With this approach we obtained approximately 60% m$^3$C32 modified tRNAs, as confirmed by hydrazine aniline cleavage [31], followed by PAGE analysis and quantitative LC-MS/MS (Fig 2A). We then quantitatively assessed the m$^3$C demethylation of these enzymatically generated substrates by quantitative LC-MS/MS. These results show that ALKBH1 reduces m$^3$C methylation levels in these tRNAs by about 25–40% (Fig 2B) and that m$^3$C32 is indeed a substrate of ALKBH1. To test whether post translational modifications of ALKBH1 that are not installed upon heterologous expression in *E. coli*, affect its activity, we also expressed and purified ALKBH1 from HEK293T cells. However, the expression system did not influence ALKBH1 activity (Fig 2B).

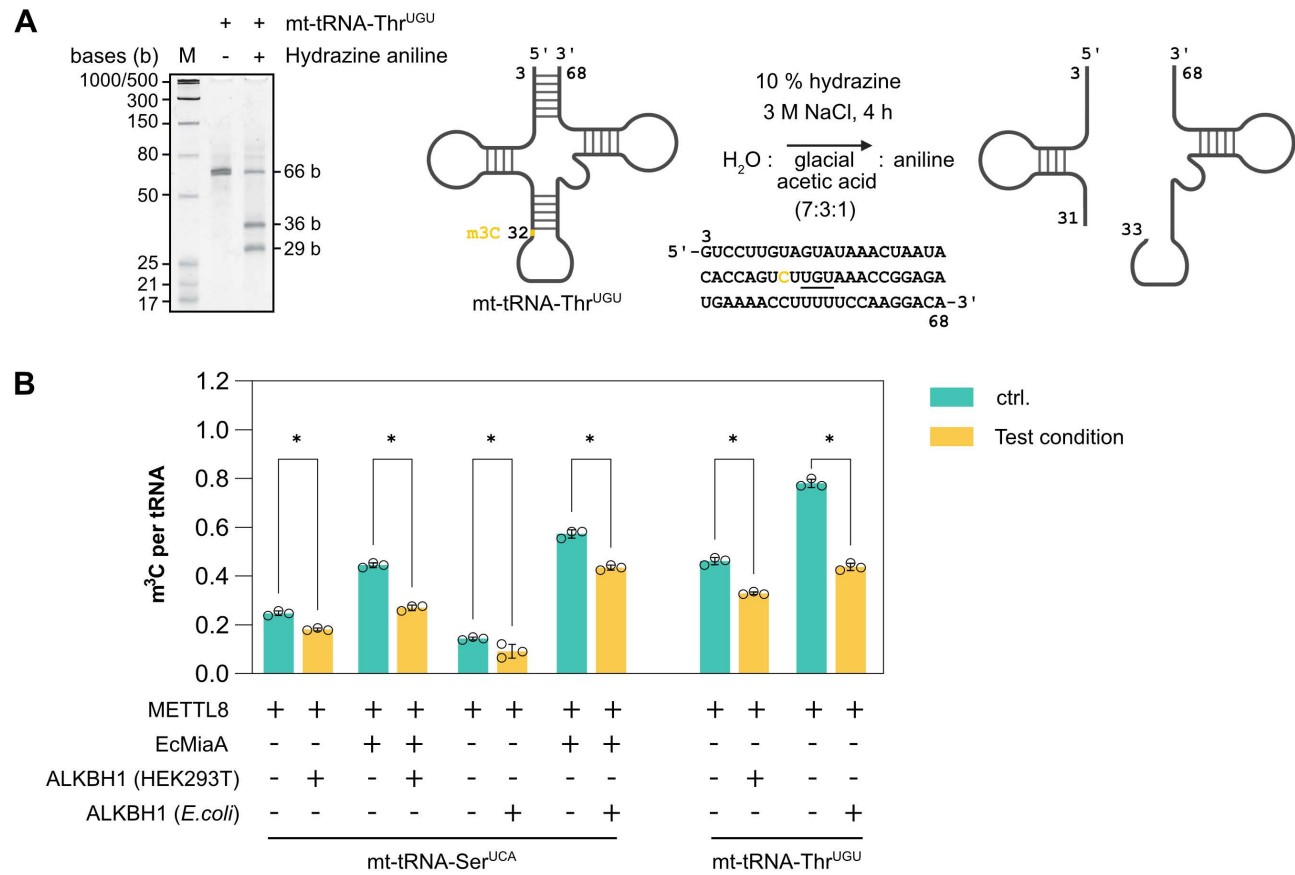

**Fig 2. *In vitro* demethylation activity of m³C in tRNA by ALKBH1. A.** Analysis of m³C modification of *in vitro* transcribed mt-tRNA-Thr^UGU by METTL8 by hydrazine/aniline cleavage and urea-PAGE. Sequence and numbering of mt-tRNA-Thr^UGU relative to wobble position 34. **B.** *In vitro* demethylation activity by ALKBH1 expressed in HEK293T or *E. coli* cells. mt-tRNA-Thr^UGU and mt-tRNA-Ser^UCA were pre-methylated at m³C32 by METTL8 mt-tRNA-Ser^UCA was additionally modified by EcMiaA (i⁶A37). Error bars in the graph represent the ± mean **S.**E. of three biological replicates. p-value mentioned in the text is calculated by unpaired t test with no correction and indicates significant differences in means (Created in BioRender. Henzeler, **B.** (2025)).

## ALKBH1 expression modulation does not impact on mt-tRNA modification level in HEK293T cells

To investigate the impact of ALKBH1 abundance on the modification of specific mitochondrial tRNAs, we overexpressed ALKBH1 or performed siRNA-mediated knock-down of ALKBH1 in HEK293T cells for 48 h (S5 Fig). It was previously shown that upon overexpression ALKBH1 localizes to mitochondria in several human cell lines [19]. Following RNA extraction, specific mitochondrial tRNAs were enriched using complementary, biotinylated DNA strands. 2 mt-tRNAs, namely mt-tRNA-Ser^UCA and mt-tRNA-Thr^UGU [23,29], carry m³C32 and may be substrates for ALKBH1. As a non-substrate control, we selected mt-tRNA-Ser^AGU which does not carry m³C32 but is modified by NSUN2 [32,33] and carries m⁵C48-C50. The known substrate, mt-tRNA-Met, was not analyzed as the ALKBH1 reaction product 5-formylcytidine (f⁵C) is too low abundant for reliable detection and quantification in our quantitative LC-MS/MS set-up. The isolated tRNAs were digested to single nucleosides, and the modified nucleosides were quantified by quantitative LC-MS/MS [34,35]. The results show that the modification abundance (including pseudouridine (Ψ), m²G, m¹A, m⁵C, m⁵U, m⁶A, t⁶A, m³C and ms²i⁶A) remained unaffected under both overexpression and knock-down conditions in HEK293T cells, grown under optimal conditions (Fig 3 and S6 Fig). One potential explanation for this unexpected finding might be that ALKBH1 has an extraordinarily long half-life which would bias knock-down experiments. We investigated the half-life of ALKBH1 using

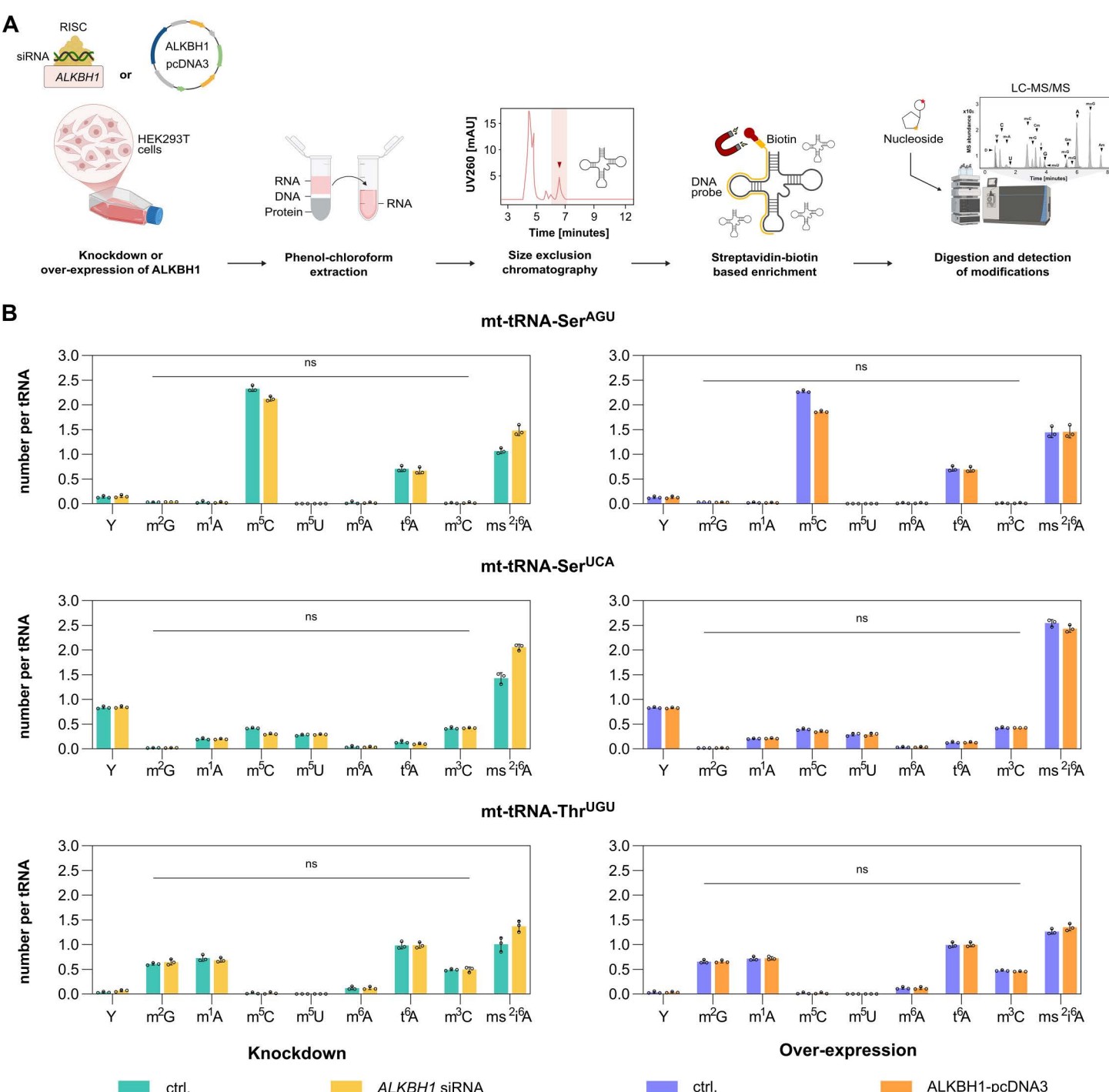

**Fig 3. ALKBH1 mediated demethylation of native mt-tRNAs in HEK293T cells. A.** Illustration of the transfection protocol for introducing plasmid or siRNA into HEK293T cells, followed by RNA extraction, enrichment and LC-MS/MS analysis. **B.** LC-MS/MS analysis of modification in transfected cells for mt-tRNA-Ser$^{AGU}$, mt-tRNA-Ser$^{UCA}$ and mt-tRNA-Thr$^{UGU}$ enrichment. Error bars in the graph represent the ± mean **S.**E. of three technical replicates. p-value mentioned in the text is calculated by unpaired t-test with no correction and indicates significant differences in mean (Created in BioRender. Henzeler, **B.** (2025)).

cycloheximide chase assay to inhibit *de novo* protein biosynthesis, which revealed that ALKBH1 abundance does not change over 72 h after addition of cycloheximide (S5 Fig), confirming its unexpected long half-life. This is also reflected in a pulse-chase nucleic acid isotope labeling coupled mass spectrometry (NAIL-MS) experiment [36], where tRNA and rRNA modifications were quantified following an 64–72 hr ALKBH1 knock-down. Again, we did not detect any significant changes in RNA modifications in the HEK293T cell line grown under optimal conditions (S7 Fig). This indicates that either remaining ALKBH1 activity is sufficient to maintain the steady-state abundance of its substrate modifications or that ALKBH1 is inactive in cells grown under non-stress conditions. We also did not observe any impact on the cell viability upon modulation (knock-down or overexpression) of ALKBH1 as assessed by the MTT assay (S8 Fig).

## Glucose concentration does not impact ALKBH1-mediated RNA-modification in HEK293T or HeLa cells

According to our previous results, ALKBH1 might act on its substrates mainly during stress conditions. Indeed, it was previously reported that glucose concentration in the growth medium affects ALKBH1 expression in HeLa cells, resulting in changes of $m^1A$ in tRNA. This leads to attenuation of translation initiation and reduced utilization of tRNAs in protein synthesis [8]. To follow up on this report, human HeLa and HEK293T cells were cultured for 8 h, as previously reported, in the presence of either 1 g/L or 4.5 g/L glucose and 2% or 10% FBS, and $m^1A$ levels in two different cytosolic tRNAs (ct-tRNA-His$^{GUG}$ and ct-tRNA-Gly$^{GCC}$) were analyzed by quantitative LC-MS/MS (Fig 4A and B). However, we did not observe variation in $m^1A$ abundance in the specific tRNAs isolated from these cell lines (Fig 4B). We also analyzed ALKBH1 -mRNA levels under these different growth conditions and observed only a minor impact from variations in glucose or FBS concentrations (Fig 4C). In addition, we carried out siRNA-mediated knock-down of ALKBH1 in HEK293T cells and cultured them in media containing 10% FBS and 4.5 g/L glucose (S9B Fig). Total tRNA was subsequently extracted and $m^1A$, $m^3C$, $m^6A$, $m^1G$ and $m^5C$ levels were quantified using quantitative LC-MS/MS. Under these conditions, we observed only minimal changes in the overall $m^1A$ and $m^1G$ abundance, with no detectable differences in $m^3C$ or $m^5C$ levels compared to control (Fig 4D). The experiment was repeated under identical conditions, but with an elongated incubation time of 24 h. We find that even 24 h of starvation conditions did change mRNA and protein levels of ALKBH1 compared to the controls (S9A and S9C Fig) Similarly, tRNA modification analysis by LC-MS/MS did not reveal any significant shifts in the overall modification patterns under varying glucose and serum conditions (S9D Fig).

## Discussion

Nucleic acid methylation is a dynamic, secondary layer essential for gene regulation. While the demethylation of 5-methylcytosine in DNA by TET enzymes can be tracked *in vivo* [20], the analogous processes in RNA remain technically challenging to observe. Here Fe(II)/α-KG-dependent dioxygenases of the AlkB-family play an important role as erasers of epitranscriptomic marks. Albeit there are numerous reports on the *in vitro* activity of these enzymes, reproduction of their *in vivo* activity appears to be challenging and possibly linked to their unclear biological roles and cellular programs.

The first report on tRNA demethylation through ALKBH1 focused on the demethylation of $m^1A$ under glucose starvation conditions and siRNA-mediated ALKBH1 knockdown. Consistent with this and other *in vitro* studies, we confirm ALKBH1 activity on $m^3C$, $m^1A$, and $m^5C$ in total tRNA extracts [8–12]. The use of defined synthetic tRNAs (with $m^3C32$ and $i^6A37$) further substantiates substrate-specific demethylation. However, such *in vitro* activities do not necessarily translate to observable changes in endogenous systems. Despite successful siRNA mediated knock-down or overexpression of ALKBH1 in human HEK293T cells, we do not observe significant changes on the nucleoside methylation levels. Our observation is also supported by a recent preprint in which CRISPR-Cas9 knock-out of ALKBH1 in 4 different cancer cell lines was shown to have no demethylation effect on tRNA modifications [37]. Similarly, we also did not observe any changes in ALKBH1-dependent RNA-modifications upon variation of the glucose or FBS concentration for 8 h in the growth media of HEK293T and HeLa cell lines, even though such effects have been previously observed [8]. Moreover, even after incubation for 24 h under these conditions, no significant changes in nucleoside modification in isolated total

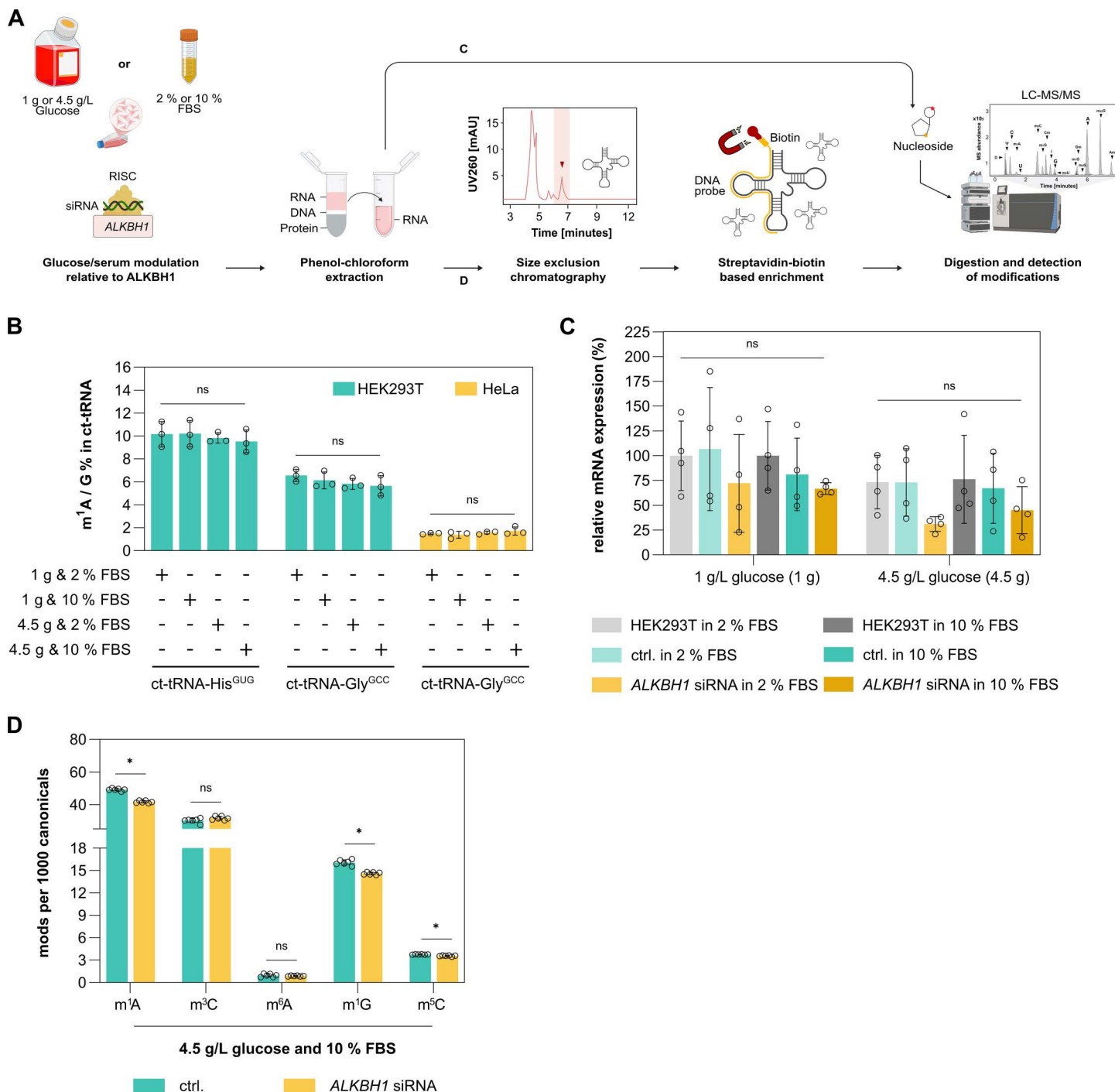

**Fig 4. Glucose-dependent demethylation activity of ALKBH1. A.** Schematic representation of glucose **(G)**- and serum (FBS)-dependent demethylation activity of ALKBH1 in HEK293T or HeLa cells. Cells were treated with either 1 g/L or 4.5 g/L glucose and 2% or 10% FBS for 8 **h.** Following treatment, total RNA was extracted and either analyzed directly by LC-MS/MS or enriched for ct-tRNA-His$^{GUG}$ and ct-tRNA-Gly$^{GCC}$, enzymatically digested, and subjected to LC-MS/MS to detect RNA modifications. **B.** LC-MS/MS analysis of m$^1$A in enriched ct-tRNAs from both HEK293T and HeLa cells grown under varying glucose and serum conditions. **C.** RT-qPCR analysis of ALKBH1 knockdown in HEK293T cells 48 h post-transfection with siRNAs targeting ALKBH1 (Negative control "ctrl": scrambled siRNA). Untransfected HEK293T cells grown under varying glucose and serum conditions were also included. **D.** LC-MS/MS analysis of RNA modifications in total RNA from siRNA-transfected HEK293T cells. Error bars in the graph represent the ± mean **S.**E. of three/six biological replicates. Statistical significance was calculated by using a one-way ANOVA without correction (Created in BioRender. Henzeler, **B.** (2025)).

tRNAs could be observed. It should be noted, however, that HEK293T cells are a highly transformed, immortalized cell line, and most other cells would be unlikely to tolerate such stress conditions over this time frame. Under these circumstances, for example during the onset of cell death, numerous factors beyond ALKBH1 may become up- or downregulated or activated or inactivated, which could also potentially influence epigenetic modifications.

Under homeostatic conditions, ALKBH1 mRNA remains stable for over 72 h, suggesting that transcriptional stability may explain the lack of functional demethylation in these cell lines und starvation conditions. Another possible explanation is that ALKBH1 is functionally engaged only under particular stress conditions, during metabolic adaptation, or in differentiated cell types. Alternatively, post-translational mechanisms, including subcellular localization or cofactor availability, may gate its catalytic activity. We also note that the use of bulk LC-MS/MS approaches limits sensitivity to localized, transcript-specific, or compartmentalized demethylation events, which may escape detection despite being biologically relevant. In sum, our data emphasize that the function of RNA demethylases such as ALKBH1 is likely highly context-dependent. For example, a recent study by Shen *et al.* demonstrated that, in acute myeloid leukemia (AML) cells, ALKBH1 facilitates codon-biased translation of rare leucine codons by converting $m^5C$ to 5-formylcytosine at the wobble position of tRNAs [16].

Even for ALKBH5, the best studied RNA eraser (target: $m^6A$ in mRNA), *in vivo* demethylation cannot be tracked in homeostatically grown cells [38,39]. These findings support the idea that the activity of RNA erasers is tightly controlled to balance the methylation density of the substrate RNAs and specific triggers might be required for their activity.

We conclude that RNA (de-)methylation reactions observed *in vitro* are not easily linked to an *in vivo* activity. In case of ALKBH1, we consider *in vivo* studies that include the LC-MS/MS characterization of both the substrate, $m^5C$, and the product, $f^5C$, as highly credible as this technology is self-validated by the interplay of product and substrate detection. Unfortunately, the oxidation of $m^1A$ by ALKBH1 leads to products that are unstable (1-hydroxymethyladenosine or 1-formyladenosine) and decompose to adenosine. Thus, *in vivo* activity cannot be cross-validated by detection of the product as adenosine is thousand-fold more abundant than $m^1A$ in tRNA. Thus, the chemistry of the products may explain why the activity of ALKBH1 towards $m^5C$ was proven *in vivo*, while it remains uncertain for $m^1A$ and $m^3C$ (that also directly decomposes to form cytidine after oxidation). In this context quantitative LC-MS/MS analysis must be considered the gold standard in nucleoside detection due to its high specificity and sensitivity, that other detection methods such as dot blots cannot provide. Through the use of NAIL in pulse-chase applications, the resolution of LC-MS/MS can be extended by the dimension of time as done in this work and our recent work on ALKBH5 [38]. Nevertheless, a main challenge for detecting and quantifying modified ribonucleosides is the reproducibility across laboratories [40]. Here variations in growth conditions, cell lines, RNA isolation and processing as well as lack of a standard form of modified RNA for researchers to use as a benchmark for validating their studies hamper the accurate quantification of RNA modifications. In addition, there is no single optimal method for all 170 known modified ribonucleosides or the more than 50 human RNA modification. We hypothesize that the highly dynamic process of RNA demethylation is only observable *in vivo* once analysis is performed at temporal resolution. We want to conclude that the absence of measurable effects in standard cell culture conditions does not negate potential physiological roles under specific stimuli or disease states. Understanding how ALKBH1 is regulated and which substrates it acts upon *in vivo* remains a central challenge for elucidating its role in RNA biology.

## Materials and methods

### Cloning, expression and purification of ALKBH1, EcMiaA and METTL8

*Homo sapiens* **ALKBH1.** For the overexpression of ALKBH1 in human cells two expression constructs were cloned in pcDNA3: one encoding the full-length ALKBH1 sequence (construct 1) and the second with the ALKBH1-sequence fused with a C-terminal Strep-tagII sequence (construct 2). For construct 1, the synthetic DNA sequence (Azenta) was cloned into pcDNA3 using the NEBuilder HiFi DNA Assembly Master Mix (New England Biolabs (NEB)) according to the manufactures protocol. This construct was used to investigate the effect of over expression of ALKBH1 on tRNA

modification levels. For construct 2 NEB Q5® Site Directed Mutagenesis Kit was used to attach a C-terminal Strep-tag II using cloning primers. The sequence was identified by sequencing. The plasmid expressing human ALKBH1 with a C-terminal Strep-tag II (construct 2) was used solely to purify ALKBH1 from human cells in order to assess whether the source organism affects enzyme activity *in vitro*. The construct was delivered to cells by forward transfection using Lipofectamine 3000 reagent (Invitrogen). 90 µg of plasmid DNA and 180 µL of Lipofectamine 3000 (Invitrogen) were diluted separately in Opti-MEM and incubated briefly. The solutions were then mixed and incubated for 15 mins at room temperature. The complex was then slowly added to a confluent T175 cell flask (HEK293T) and incubated at 37°C under 5% $CO_2$ for 48 h. Post incubation, cells were harvested and resuspended in 7.5 mL lysis buffer (100 mM Tris-HCl, 150 mM NaCl, TCEP, pH 8) supplemented with DNaseI (AppliChem) and protease inhibitors (Roche). and lysed by sonication using Sonic Dismembrator 120 (Fisherbrand) and the lysate was cleared by centrifugation (20,000 x g; 30 min; 4°C). Following this, nucleic acids were removed from the lysate by addition of polyethylenimine (PEI) dropwise (Merck) to a final concentration of 0.5% w/v followed by a second centrifugation step (20,000 x g; 20 min; 4° C). The cleared lysate was filtered and incubated with 1 mLStrepXT beads (IBA Lifesciences), at 4°C for 30 min. The incubated resin was loaded into a column and washed with wash buffer (100 mM Tris-HCl, 150 mM NaCl, 1 mM TCEP, pH 8.0) and the protein was eluted with elution buffer (100 mM Tris-HCl, 150 mM NaCl, 1 mM Tris(2-carboxyethyl)phosphine (TCEP), 50 mM biotin, pH 8.0). ALKBH1 containing fractions were pooled and concentrated with an Amicon Ultracell Centrifugal filter unit (MWCO 10 kDa, Merck) and flash-frozen in liquid nitrogen and stored at −80° C for further use.

For expression in *E. coli*, a synthetic, codon-optimized sequence encoding human ALKBH1[(19-369)], fused in-frame with an N-terminal Strep-affinity tag followed by a Tabacco Etch Virus (TEV) protease cleavage site (Azenta Life Science), was cloned into the pET28 vector (Novagen). The identity of the construct was verified by DNA sequencing. For protein expression, the plasmid was transformed into *E. coli* BL21(DE3) cells (Novagen). An overnight culture was prepared in LB medium supplemented with 50 µg/mL kanamycin. The following day, the culture was diluted 1:100 into fresh 2×YT medium containing 50 µg/mL kanamycin and incubated at 37°C with shaking at 180 rpm until an $OD_{600}$ of 0.8 was reached. Protein expression was induced with 0.5 mM isopropyl-β-D-1-thiogalactopyranoside (IPTG) and incubated overnight at 18°C and subsequently harvested by centrifugation (4000×g; 20 min, RT). The cell pellet was resuspended in 30 mL lysis buffer (50 mM Tris-HCl, 300 mM NaCl, 1 mM TCEP 0.5% v/v N-lauroylsarcosine, pH 8) supplemented with 250 U/ µL Benzonase (Merck), Lysozyme and protease inhibitors (Roche). Cells were then lysed by homogenization using an EmulsiFlexC5 (Avestin Inc.) and the lysate was cleared by centrifugation (22,000×g; 30 min; 4°C). The cleared lysate was filtered and loaded onto 1 mL StrepXT beads (IBA Lifesciences), the column was then washed with wash buffer (100 mM Tris-HCl pH 8.0, 150 mM NaCl, 1 mM TCEP) and the protein was eluted with elution buffer (100 mM Tris-HCl pH 8.0, 150 mM NaCl, 1 mM TCEP, 50 mM biotin). Fractions containing ALKBH1 were pooled and His-tagged TEV-protease was added (1:10) and incubated overnight. TEV-protease was then removed using Ni-NTA beads (Qiagen) and tag-free ALKBH1 was obtained in the flow-through and diluted in CaptoQ-binding buffer (50 mM Tris/HCl pH 8.7, 1 mM TCEP). The diluted protein was then loaded onto a 1 mL CaptoQ column in a ÄktaGo FPLC system (Cytiva) and eluted with elution buffer (50 mM Tris/HCl pH 8.7, 1 M NaCl, 1 mM TCEP) with a gradient to 70% elution buffer over 10 column volumes (CV). ALKBH1 containing fractions were pooled and concentrated with an Amicon Ultracell Centrifugal filter unit (MWCO 10 kDa, Merck) and loaded onto a Superdex Increase 75 10/300 column (Cytiva), equilibrated with 50 mM Tris/HCl pH 7.5, 300 mM NaCl, 1 mM TCEP, 5% v/v glycerol. Finally, the fractions containing pure ALKBH1 were pooled, concentrated, flash-frozen in liquid nitrogen and stored at −80°C for further use.

***Escherichia coli* MiaA.** The full-length sequence encoding *E. coli* MiaA (Uniprot No. 16384) was PCR-amplified from genomic *E. coli* K12 DNA and cloned in frame with a C-terminal His_6-tag in pET28 via restriction digestion and ligation, using the NcoI and XhoI restriction sites and sequenced for identity. For protein expression the plasmid was transformed into T7express (NEB) and cultured overnight in LB medium supplemented with 50 µg/mL Kanamycin. Post incubation the culture was diluted 1:100 in 2YT medium (50 µg/mL kanamycin) and incubated at 37°C, 180 rpm until an $OD_{600}$ of 0.8 was

reached. Protein expression was induced with 1 mM IPTG and incubated overnight at 16°C and subsequently harvested by centrifugation (4000×g; 20 min, RT). The cell pellet was resuspended in 20 mL lysis buffer (50 mM Tris-HCl, 300 mM NaCl, 20% v/v glycerol, 20 mM imidazole, pH 8) supplemented with DNase I (AppliChem) and protease inhibitors (Roche). Cells were homogenized using an EmulsiFlexC5 (Avestin Inc.) and the lysate was cleared by centrifugation (22,000×g; 30 min; 4°C). The cleared lysate was filtered and loaded onto a 5 mL HisTrap crude column (Cytiva) on an ÄktaGo FPLC (Cytiva) at 8°C, the column washed with lysis buffer and the protein was eluted with elution buffer (50 mM Tris-HCl, 300 mM NaCl, 20% v/v glycerol, 300 mM Imidazole, pH 8). Fractions containing the pure protein were pooled, concentrated with an Amicon Ultracell Centrifugal filter unit (MWCO 10 kDa, Merck Millipore) and was exchanged to storage buffer (50 mM Tris-HCl, 150 mM NaCl, 2 mM dithiotreitol (DTT) pH 8) using a PD10 column (Cytiva). The protein (yield ~38 mg/L expression culture) was aliquoted, flash-frozen in liquid nitrogen and stored at −80°C.

*Homo sapiens* METTL8. The synthetic sequence encoding for the human METTL8 isoform1 Δ1–22 (Uniprot No. B3KW44, Azenta Life Science) Δ1–22 was cloned in pET28 in frame with a C-terminal His$_6$-tag using the restriction sites NcoI and XhoI and sequenced for identity verification. For protein expression the plasmid was transformed into T7express (NEB) and cultured overnight in LB medium supplemented with 50 µg/mL Kanamycin. The following day, the culture was diluted 1:100 in 2YT medium (50 µg/mL kanamycin) and incubated at 37°C, 180 rpm until an $OD_{600}$ of 0.8 was reached. Protein expression was induced with 0.5 mM IPTG and incubated overnight at 18° C and subsequently harvested by centrifugation (4000×g; 20 min, 4°C). The cell pellet from a liter expression culture was resuspended in 25 mL lysis buffer (50 mM Tris-HCl, 500 mM NaCl, 5% v/v glycerol, 10 mM imidazole, pH 8) supplemented with DNase I (AppliChem) and protease inhibitors (Roche). Cells were homogenized using an EmulsiFlexC5 (Avestin Inc.) and the lysate was cleared by centrifugation (20,000×g; 30 min; 4°C). Nucleic acids were removed from the lysate by adding PEI dropwise to a final concentration of 0.5% w/v followed by a second centrifugation step (20,000 x g; 20 min; 4°C). The cleared lysate was filtered and incubated with Ni-NTA resin at 4°C for 30 min. The incubated resin was loaded into a column and washed 5 times with 2 CV of wash buffer (50 mM Tris-HCl, 1 M NaCl, 5% v/v glycerol, 20 mM imidazole, pH 8). Afterwards, the His-tagged protein was eluted with elution buffer (50 mM Tris-HCl, 300 mM NaCl, 20% v/v glycerol, 300 mM imidazole, pH 8). The collected elution fractions were combined and diluted 1:10 and loaded onto a HiTrap™ Heparin HP 5 mL column (Cytiva). The column was then washed with binding buffer (50 mM Tris-HCl, 30 mM NaCl, 30 mM imidazol, 5% glycerol, pH 8) and the protein was eluted with elution buffer (20 mM Tris-HCl, 1 M NaCl, 5% v/v glycerol, pH 8) in a continuous gradient. Fractions containing pure protein were pooled, concentrated with an Amicon Ultracell Centrifugal filter unit (MWCO 10 kDa, Merck Millipore) and the buffer was exchanged to storage buffer (50 mM Tris-HCl, 150 mM NaCl, 2 mM DTT, pH 8). The protein was aliquoted, flash-frozen in liquid nitrogen and stored at −80° C.

### *In vitro* transcription of tRNA

Synthetic DNA strands (biomers.net GmbH) encoding for the sequences of human tRNAs and a 5' T7 RNA polymerase promoter sequence, were hybridized and used as template. For the *in vitro* transcription the reaction was set up (NEB, HiScribe T7 Quick High Yield RNA Synthesis Kit) according to the manufactures' protocol, followed by removal of the dsDNA template using DNaseI. The RNA products were then purified (NEB Monarch RNA Cleanup Kit) and stored at −80° C. To ensure proper folding, the tRNA was heated to 85° C and subsequently cooled down to 20°C over 30 min.

### Dynamic light scattering (DLS)

*In vitro*-transcribed tRNA was diluted to 1 µg/µL in 50 µL of buffer (20 mM Tris-HCl, pH 7.5; 10 mM MgCl$_2$; 100 mM NaCl), heated to 85 °C, and either slowly cooled to room temperature or immediately placed on ice prior to DLS analysis (Wyatt DynaPro NanoStar). In addition, tRNA was diluted in 10 mM Tris-HCl (pH 7.5) and 10 mM EDTA containing 55% (v/v) for-mamide. Samples were diluted to 200 ng/µL prior to dynamic light scattering analysis, which consisted of 20 consecutive measurements per sample.

## Methylation assay

A 50 µL reaction mixture containing 50 mM Tris-HCl (pH 7.5), 20 mM KCl, 10 mM $MgCl_2$, 1 mM DTT, 1 mM spermidine, 200 µM S-adenosyl methionine (SAM), 10 µM mt-tRNA-Thr[UGU] or mt-tRNA-Ser[UCA] and 15–30 µM METTL8 was incubated at 37° C for 1 h. The resulting samples were purified using either the Monarch Spin RNA Cleanup Kit (10 µg NEB) according to the manufacturer's instructions or ethanol precipitation. For ethanol precipitation, samples were mixed with 25 µL of 3 M sodium acetate (pH 5.5) and 0.75 mL of 100% ethanol, snap-frozen in liquid nitrogen, and centrifuged at 16,000 x g for 40 min at 4°C. The pellet was washed twice with 0.7 mL of 70% ethanol and resuspended in 25 µL of water. To ensure proper folding the tRNA was heated to 85°C and subsequently cooled down to 20°C over 30 min.

## ALKBH1 *in vitro* activity

Recombinant ALKBH1 protein was diluted as needed using its designated storage buffer. For each *in vitro* reaction, the final buffer composition was adjusted to include: 50 mM HEPES (pH 7.5), 15 mM KCl, 2 mM L-ascorbate, 2 mM $MgCl_2$, 300 µM α-ketoglutarate (α-KG), and 105 or 300 µM ferrous ammonium sulfate [$Fe(II)(NH_4)_2(SO_4)_2$]. To ensure reproducibility and stability, fresh aqueous stock solutions were prepared in either 10 mL or 50 mL centrifuge tubes at the following concentrations: 1 M HEPES (pH 7.5), 100 mM KCl, 20 mM L-ascorbate, 20 mM $MgCl_2$, 10 mM α-ketoglutarate, and 10 mM [$Fe(II)(NH_4)_2(SO_4)_2$]. Notably, the L-ascorbate, α-KG, and Fe(II) solutions were freshly prepared immediately before each experiment to prevent oxidation and ensure activity. Where feasible, a master mix of the salt components was prepared to reduce pipetting errors and maintain consistency across reactions. Each 50 µL reaction mixture consisted of the appropriate buffer components, ALKBH1 enzyme, and RNA substrate. The mixture was gently homogenized by pipetting up and down and subsequently incubated in a heating block at 37° C for 2 h. The enzymatic reaction was halted, and RNA was precipitated by the addition of 500 µL (10 × volume) of cold 2% (w/v) lithium perchlorate in acetone. After 5 min incubation at room temperature, the mixture was centrifuged at 5,000 × g for 5 min to pellet the RNA. The supernatant was carefully removed, and the RNA pellet was air-dried briefly before resuspension in 20 µL of ultrapure water. Following resuspension, the RNA was enzymatically digested for downstream analysis.

## Nucleic acid digestion for quantitative nucleoside analysis

RNA samples were enzymatically digested to nucleosides using a defined master mix (See S2 Table) containing buffer components, nucleases, phosphatase, and protective agents. The digestion buffer system consisted of Tris-HCl and $MgCl_2$, which served as the primary buffering and divalent cation components, respectively. Nuclease digestion was carried out using Benzonase (endonuclease) and snake venom phosphodiesterase (PDE1), while dephosphorylation of nucleotides was achieved using calf intestinal phosphatase (CIP). To prevent chemical modifications of the released nucleosides during digestion, the reaction mixture was supplemented with pentostatin and tetrahydrouridine (THU) to inhibit deamination, and butylhydroxytoluene (BHT) as an antioxidant. To account for possible contamination of the THU stock solution with dihydrouracil (D), as a control samples without added THU were also prepared and, following the quantification, compared. Each digestion reaction contained 20 µL of RNA sample mixed with 10 µL of freshly prepared master mix. For RNA inputs differing from the standard 4 µg, the volumes of Tris-HCl and $MgCl_2$ in the master mix were proportionally adjusted. Unless otherwise stated, all procedures refer to the digestion of 4 µg of RNA. After mixing thoroughly, the samples were incubated at 37°C for 2 h. Following digestion, each reaction was diluted with half the volume of LC-MS-compatible buffer. For LC-MS quantification, 1 µL of a 10 × SILIS (Stable Isotope-Labeled Internal Standard) solution was co-injected with each sample [41]. SILIS digestion for up to 3 µg of RNA was conducted using a protocol designed for 10 µg input RNA. In optional experiments, 0.1 × volumes of 100 µM theophylline and additional LC-MS/QQQ buffer were included for modulation. Final buffer volumes were adjusted to reach a target nucleosides concentration of 20 ng/µL prior to LC-MS/MS analysis.

## Synthesis of m³C containing RNA

Synthesis of the m³C phosphoamidite and incorporation into RNA was carried out as previously reported [42,43], with a slight adaption of the RNA oligonucleotide synthesis. Coupling was done using Activator 42 (0.25 M) in acetonitrile and 5'-detritylation was achieved using 3% dichloroacetic acid in dichloromethane, on a ABI Oligo DNA RNA Synthesizer. The identity of the products was confirmed by NMR, ESI- or MALDI-MS.

## Quantitative LC-MS analysis

Quantitative LC-MS analysis was performed using an Agilent Infinity 1290 HPLC system coupled to an Agilent G6490 or G6470A Triple Quadrupole mass spectrometer equipped with an electrospray ionization source. The Agilent G6490 was operated in positive ion mode with the following parameters: cell accelerating voltage of 5 V, nitrogen ($N_2$) gas temperature of 120°C with a flow rate of 11 L/min, sheath gas ($N_2$) temperature of 280° C at 11 L/min, capillary voltage of 3000 V, nozzle voltage of 0 V, and nebulizer pressure of 60 psi. High-pressure radiofrequency was set to 100 V, while low-pressure radio-frequency was set to 60 V. the Agilent G6470A was operated in positive ion mode with the following parameters: cell accelerating voltage of 5 V, nitrogen ($N_2$) gas temperature of 230° C with a flow rate of 6 L/min, sheath gas ($N_2$) temperature of 400°C at 12 L/min, capillary voltage of 2500 V, nozzle voltage of 0 V, and nebulizer pressure of 40 psi. High-pressure radiofrequency was set to 100 V, while low-pressure radiofrequency was set to 60 V. Data acquisition was conducted in dynamic multiple reaction monitoring (DMRM) mode. Chromatographic separation when using the Agilent G6490 was achieved using an Uptisphere C18-HDO column (3.0 µm, 150×2.1 mm; Interchim, UP3HDO-150/021) at 35°C with a flow rate of 0.35 mL/min. The mobile phase consisted of 5 mM ammonium acetate ($NH_4OAc$, pH 4.9) as aqueous buffer A and 2 mM ammonium formate ($NH_4COOH$) in 80% acetonitrile as organic buffer B. The gradient program was as follows: 100% A for 0.5 min, followed by an increase to 10% B over 5.5 min, then to 20% B from 6.0 to 8.5 min, followed by a ramp to 80% B in 1 min, held for 1.5 min, and re-equilibrated to 100% A over 0.5 min with a re-equilibration period of 2.2 min. Alternatively, when using the Agilent G6470A, a Synergi, 2.5 µm Fusion-RP, 100 Å, 100 mm x 2 mm column (Phenomenex, Torrance, CA, USA) was used with a flow rate of 0.35 mL/min. Here the mobile phase consisted of 5 mM ammonium acetate ($NH_4OAc$, pH 5.3) as aqueous buffer A and 100% acetonitrile as buffer B. The gradient program was as follows: 100% A for 1 min, followed by an increase to 10% B over a period of 4 min, then to 40% B from 6.0 to 8 min and maintained for 1 min, before returning to 100% A. Each sample was analyzed by co-injecting a 10 µL aliquot with 1 µL of a stable isotope-labeled internal standard (ISTD). Data was processed using Agilent MassHunter quantitative analysis software with integrated calibration. Calibration curves ranged from 0.0488 to 100 pmol for canonical nucleosides and 0.0024 to 5 pmol for modified nucleosides (12 calibration levels, 1:2 serial dilution). The modified nucleosides were quantitatively related to the molar amount of each canonical nucleoside which were normalized to the expected occurrence in the respective tRNA sequence. DMRM transitions and retention times (RT) were as follows: m3C: 258.1→126.0 (RT 4.2 min); m³C-ISTD: 261.1→129.0 (RT 4.2 min).

## Data analysis of nucleoside LC–MS/MS

Raw data were analyzed using quantitative and qualitative MassHunter Software from Agilent. The signals for each nucleoside from dynamic multiple reaction monitoring (DMRM) acquisition were integrated along with the respective SILIS. The signal areas of nucleoside and respective SILIS were set into relation to calculate the nucleoside isotope factor (NIF):

$$NIF = \frac{signal\ area\ (nucleoside)}{signal\ area\ (SILIS)}$$

The NIF was then plotted against the molar amount of each calibration, and regression curves were plotted through the data points. The slopes represent the respective relative response factors for the nucleosides (rRFN) and enable absolute quantification of nucleosides. Calibration curves were plotted automatically by quantitative MassHunter software from

Agilent. Molar amounts of nucleosides in samples were calculated using the signal areas of the target compounds and SILIS in the samples and the respective rRFN, determined by calibration measurements. This step was also done automatically by quantitative MassHunter software. The detailed calculation is depicted in the following equation:

$$n_{\text{sample nucleoside}} = \frac{\text{signal area}_{\text{sample nucleoside}}}{\text{rRFN}_{\text{nucleoside}} \times \text{signal area}_{\text{respective SILIS}}}$$

The molar amount of modified nucleosides was then normalized to respective tRNA population to calculate the amount of modification per tRNA. This was done using the expected amount of canonical nucleosides of the respective tRNA population.

$$n_{\text{tRNA}} = \frac{\frac{n_C}{\#C} + \frac{n_U}{\#U} + \frac{n_G}{\#G} + \frac{n_A}{\#A}}{4}$$

$$\text{number of modifications per tRNA} = \frac{n_{\text{modifications}}}{n_{\text{tRNA}}}$$

The molar amount of modified nucleosides in total RNA and total tRNA was normalized to the molar amount of 1000 canonical nucleosides:

$$\frac{\#modification}{1000\ nts} = \frac{n_{\text{modification}}}{\left( \frac{n_G}{1000} + \frac{n_A}{1000} + \frac{n_C}{1000} + \frac{n_U}{1000} \right)}$$

For experiments including nucleic acid isotope labeling, the isotopologues were normalized to the labeled canonical nucleosides to differentiate between pre-existing modifications and new modifications in the respective tRNA transcripts.

**Cell Lines**

HEK293T/HeLa cells were grown and maintained in Dulbecco's Modified Eagle's Medium (DMEM) supplemented with 10% fetal bovine serum, 1% penicillin-streptomycin, and 1% L-glutamine (Life Technologies) at 37°C and under 5% $CO_2$ atmosphere.

**RNA Extraction**

Cells were lysed in TRI Reagent and stored at −80°C until further analysis. After defrosting, 0.2 mL of chloroform was added to each sample. The samples were shaken vigorously for 15 sec and allowed to stand at room temperature for 5 min. The mixtures were then centrifuged at 12,000 × g for 10 min at 4°C to separate the phases. The upper aqueous phase was mixed with isopropanol in a 1:1 ratio to precipitate total RNA. After overnight incubation at −20° C, samples were centrifuged at 12,000 × g for 30 min at 4°C. The supernatant was removed, and RNA pellets were washed twice with 150 µL 70% ethanol and centrifuged at 12,000 × g for 10 min at 4° C. After removing the supernatant, the samples were placed on the bench for 10 min to let the remaining ethanol evaporate. The RNA was then resuspended in 50 µL of Milli-Q water, and the concentration of each sample was measured using a nanophotometer (Biozym).

**RNA isolation using Size-Exclusion Chromatography (SEC)**

SEC was performed on an Agilent HPLC 1100 Series (Agilent Technologies, Santa Clara, CA, USA) equipped with an AdvanceBio SEC 300 Å, 2.7 µm, 7.8 × 300 mm for tRNA purification using 0.1 M ammonium acetate buffer (pH 7) at a

flow rate of 1 mL/min and a temperature of 40°C. To determine the retention time of tRNA, a small amount of total RNA extracted from HEK293T cells was first injected as a test sample. The tRNA eluted between 6.85 and 8.0 min. Based on this, the start and end times were defined in the chromatogram for the fraction collector to isolate tRNA from the actual sample. The excess solvent from the collected tRNA fraction was reduced to 50 µL using a Savant SpeedVac SPD120 (Thermo Fisher Scientific, Waltham, MA, USA). tRNA from samples were precipitated by adding 0.1 × volume of 5 M ammonium acetate (final concentration: 0.5 M) and 2.5 x volume of 100% ethanol (final concentration: 70% v/v). The samples were incubated overnight at −20° C, followed by centrifugation at 12,000 × g for 1 h at 4°C. After removing the supernatant, the RNA pellet was left to air-dry at room temperature. The RNA was then resuspended in 30 µL of Milli-Q water, and the concentration of each sample was measured using a Nanophotometer.

### Reverse transcription quantitative real-time PCR (RT-qPCR)

Universal SYBR Green Select Master Mix (Life Technologies) and custom designed primers were used to perform quantitative real-time PCR on qTower3 (Analytik Jena) system. Custom designed primers were purchased from Sigma-Aldrich for ALKBH1, β-actin and 18S rRNA. The levels of β-actin mRNA and/or 18S rRNA were used as a reference for normalizing the levels of target gene mRNAs. The data presented were normalized against β-actin and the fold or percentage (%) change in the expression levels of target gene were computed using relative $2^{-\Delta\Delta CT}$ method or absolute quantification.

### siRNA-mediated transient knockdown

Pre-designed small-interfering RNAs (siRNAs) targeting human ALKBH1 (Thermo Fisher Scientific) and a non-targeting control siRNA (Silencer™ Negative Control No. 1, Thermo Fisher Scientific) were used for RNA interference experiments. Transfections were performed using either Lipofectamine™ RNAiMAX (Thermo Fisher Scientific) or jetPRIME® (Sartorius) transfection reagent, following the manufacturer's protocol with minor optimizations. For Lipofectamine RNAiMAX-mediated transfection, siRNAs were diluted to a final concentration of 25 nM in Opti-MEM™ I Reduced Serum Medium (Thermo Fisher Scientific). Separately, 12 µL of Lipofectamine RNAiMAX reagent was also diluted in Opti-MEM™. The diluted siRNA and reagent solutions were gently mixed and incubated briefly at room temperature to allow complex formation. The siRNA-lipo complexes were then combined with HEK293T cell suspensions or directly added to adherent cells in monolayer cultures. Cells were incubated at 37°C in a humidified incubator with 5% $CO_2$ for 48 h before further analysis. For experiments involving esiRNA (endoribonuclease-prepared siRNA), HEK293T cells were seeded in T25 flasks containing 5 mL of unlabelled NAIL-MS medium. Transfections were carried out 24 h post-seeding, at approximately 30% confluency. A transfection mix (250 µL) was prepared by first mixing esiRNA with jetPRIME® buffer, followed by 10 s of vortexing. The jetPRIME® reagent was then added, mixed briefly (vortexed for 1 s), and incubated for 10 min at room temperature. The resulting transfection mix was added to the culture flask dropwise, evenly distributed across the surface, and gently swirled to ensure homogeneous delivery. Cells were incubated for 48 h under standard conditions before downstream processing.

### Overexpression of ALKBH1 in HEK293T cells

Plasmid expressing human ALKBH1 was delivered to cells by forward transfection using Lipofectamine 2000 reagent (Invitrogen). 4 µg of plasmid DNA and 12 µL of Lipofectamine 2000 (Invitrogen) were diluted separately in Opti-MEM and incubated briefly. The solutions were then mixed and incubated for 30 min at room temperature. The complex was then mixed with cell suspension before seeding in a cell culture plate or added to cell monolayers and incubated at 37°C under 5% $CO_2$ for 48 h before further processing.

### Isolation of tRNA isoacceptors

Specific RNA molecules were purified using oligonucleotide (ON) hybridization [35]. A 30–40 nucleotide oligonucleotide, reverse-complementary to the sequence of the target RNA, to which a biotin tag was attached via a short, optional AAA

adapter, was designed for each tRNA molecule to be purified. For mitochondrial tRNA isoacceptor purification, 3 μg of total tRNA was mixed with 100 pmol reverse complementary, biotinylated DNA oligonucleotides in a total volume of 100 μL of 5 x SSC buffer (0.75 M NaCl, 75 mM trisodium citrate, pH 7). The sequences of DNA oligonucleotide were taken from a previous publication [30]. The sequences can be found in S6 Table. The mixture was incubated at 90° C for 3 min for denaturation followed by a hybridization step at 65° C for 10 min. For each sample, 25 μL magnetic Dynabeads® Myone™ Streptavidin T1 (Thermo Fisher Scientific, Waltham, MA, USA) were primed three times using Bind and Wash buffer (B&W, 5 mM Tris-HCl, pH 7.5, 0.5 M EDTA, 1 M NaCl) and once using 5 x SSC buffer. An aliquot of 25 μL magnetic beads in 5 x SSC buffer was added to each sample and incubated at RT for 30 min at 600 rpm. Magnetic racks were used to separate the beads from unbound tRNA and the magnetic beads were washed once using 50 μL of 1×SSC buffer and three times using 25 μL of 0.1×SSC buffer. Elution of the desired tRNA was carried out in 15 μL water at 75° C for 3 min. Samples were directly used for LC-MS preparations. Each isoacceptorwas purified as triplicate.

## Immunoblotting

Cell lysates for blotting were prepared in a lysis buffer (250 mM NaCl, 62.5 mM Tris-HCl pH 7.4, 0.5% sodium dodecyl sulfate, 0.5% sodium deoxycholate, 1% Triton X-100) supplemented with protease inhibitor cocktail (Roche). The cell lysates were heat shocked at 96°C for 10 min followed by cooling. Using Pierce's BCA protein assay kit, total protein was quantified following the manufacturer's protocol. Equal amounts of protein were resolved in sodium dodecyl sulfate – poly-acrylamide gel electrophoresis (SDS-PAGE), transferred to a nitrocellulose membrane blocked with 5% (w/v) milk or 3% (w/v) bovine serum albumin (BSA) and incubated with primary antibody overnight at 4°C. Following a wash, the blots were treated with the secondary antibody for 1 h. The blots were re-probed by stripping with 0.5 M sodium hydroxide for 8 min at room temperature. Western blotting detection reagents SuperSignal™ West Pico PLUS Chemiluminescent Substrate were used to visualize the protein bands, and imagery was captured on an Amersham ImageQuant 800 imaging system. Images were exported as TIFFs and compiled into figures using Affinity Designer 2/ Inkscape. In cases where brightness and/or contrast adjustments were required, the changes were applied to the entire image. Using Amersham ImageQuant 800, protein band intensities were measured and normalized.

## Cycloheximide chase assay

A549 cells were treated with mock (DMSO only) or drug (Cycloheximide 300 μg/mL in DMSO). Cells were harvested at different time points and total cell lysates were prepared using the lysis buffer (Immunoblotting), and the ALKBH1 and GAPDH polypeptides were detected by Western blotting. ALKBH1 and GAPDH bands were quantified using Amersham ImageQuant 800. The level of ALKBH1 in each sample was then normalized against the level of GAPDH in the sample. Finally, the normalized levels of ALKBH1 in mock samples were considered 100% to compare its level to ALKBH1 knock-down cells

## Supporting information

**S1 Fig. SDS-PAGE analysis of purified ALKBH1, METTL8 and EcMiaA. A.** Human ALKBH1 and ALKBH1[I218A], ALK-BH1[AxA], METTL8 and *E. coli* MiaA were overexpressed and purified from *E. coli*. **B.** Human ALKBH1 was overexpressed and purified from HEK293T cells.
(TIF)

**S2 Fig. Analysis of the *in vitro* ALKBH1 activity using the Succinate-Glo™ JmjC Demethylase/Hydroxylase Assay (Promega).** Activity of Fe(II)/α-KG dependent dioxygenases result in the conversion of α-KG to succinate, which in the assay is first converted to ATP that is finally measured by a luciferase/luciferin reaction. The light generated correlates to the amounts of succinate produced by the dioxygenases as a measure for the dioxygenase activity. As substrates

either total tRNA extracted from HEK293T cells, as well as a synthetic RNA oligonucleotide as mimic for the mitochondrial tRNA$^{Ser}$ anticodon arm, with the single m$^3$C32 modification were used. Both were incubated either with 1 µM or 5 µM of wildtype ALKBH1, ALKBH1$^{I218}$, which cannot undergo auto hydroxylation, or inactive ALKBH1$^{AxA}$ as a control. This shows that the assay is also sensitive to auto-hydroxylation, since overall lower amounts of succinate are generated when ALKBH1$^{I218}$ is added to total tRNAs, compared to the wild type (left panel). Moreover, the synthetic tRNA-anticodon arm mimic seems not to be a suitable substrate for ALKBH1, since no changes in succinate levels can be observed when ALKBH1$^{I218}$, which cannot undergo auto hydroxylation was added (right panel). Without addition of α-KG to ALKBH1 no signal is observed.
(TIF)

**S3 Fig. Analysis of *in vitro* transcribed tRNAs.** A) Dynamic light scattering (DLS) analysis of folding of *in vitro* transcribed mt-tRNA-Thr$^{UGU}$. Denaturing by heat and slow cooling results in monodispersed sample, indicating homogenous and correct folding. Heating followed by rapid cooling on ice leads to aggregation that can be seen by polydispersed distribution. Polydispersity of the sample is also increased by addition of formamide. B) Urea-PAGE (8%) analysis of *in vitro* transcribed RNA of mt-tRNAs. Both tRNAs were treated with METTL8. Mt-tRNA-Ser$^{UCA}$ was first treated with EcMiaA.
(TIF)

**S4 Fig. Quantification of RNA modifications in *in vitro* transcribed ct-tRNA-Val$^{AAC}$ following ALKBH1 directed demethylation, with and without methyl methanesulfonate (MMS) treatment.** A – E. Absolute levels of m$^1$A (A), m$^5$C (B), m$^3$C (C), m$^7$G (D) and m$^1$G (E) in *in vitro* transcribed ct-tRNA-Val$^{AAC}$ with and without MMS treatment (w/o MMS). Data represents n = 3 biological replicates. Error bars in graph represent the ± mean S.E. of three biological replicates. p-value (p > 0.05 = ns, p ≤ 0.05 = *, p ≤ 0.01 = **, p ≤ 0.001 = *** and p ≤ 0.0001 = ****) mentioned in the text is calculated by one-way ANOVA and indicates significant differences in medians.
(TIF)

**S5 Fig. Expression of ALKBH1 and analysis of ALKBH1 half-life. An**alysis of *in vitro* transcribed tRNAs. A) Dynamic light scattering (DLS) analysis of folding of *in vitro* transcribed mt-tRNA-Thr$^{UGU}$. Denaturing by heat and slow cooling results in monodispersed sample, indicating homogenous and correct folding. Heating followed by rapid cooling on ice leads to aggregation that can be seen by polydispersed distribution. Polydispersity of the sample is also increased by addition of formamide. B) Urea-PAGE (8%) analysis of *in vitro* transcribed RNA of mt-tRNAs. Both tRNAs were treated with METTL8. Mt-tRNA-Ser$^{UCA}$ was first treated with EcMiaA.
(TIF)

**S6 Fig. Quantification of total tRNA modifications following ALKBH1 knockdown or over expression.** A-D. LC-MS/MS analysis with absolute levels for Y, D, m$^2$G, m$^1$A, m$^5$C, m$^{22}$G, G$_m$ and C$_m$ (A). Absolute levels for m$^7$G, m$^5$U, m$^6$A, m$^1$G, I, i$^6$A, t$^6$A and m$^1$I (B). Absolute levels for A$_m$, Q, ManQ, GalQ, U$_m$, acp$^3$U, m$^3$C and ac$^4$C (C). Absolute levels for m$^3$U, mcm$^5$U, ncm$^5$U, mcm$^5$s$^2$U, m$^{6,6}$A, m$^5$U$_m$, m$^6$t$^6$A and ms$^2$i$^6$A (D). Error bars in the graph represent the ± mean S.E. of six biological replicates. p-value mentioned in the text is calculated by unpaired t-test with no correction and indicates significant differences in median.
(TIF)

**S7 Fig. Quantification of RNA modifications in the pulse-chase NAIL-MS following ALKBH1 knockdown.** A-E. Absolute levels for m$^1$A (A), m$^5$C (B), m$^6$A (C), m$^3$C (D) and m$^7$G (E) per respective RNA molecule (left column: 28S rRNA; middle column: 18S rRNA; right column: tRNA). Each boxplot is divided into three bars: the left column shows the old, unlabeled nucleosides before medium change (t = 0 h, 64 h after seeding), the middle bar shows the old, unlabeled nucleosides after incubation with stable isotope-labeled medium (t = 8 h, 72 h after seeding), and the right bar shows the new,

labeled nucleosides after 8 h* of incubation. At each time point, RNA was isolated from siRNAs targeting ALKBH1 (Negative control "ctrl": scrambled siRNA). All values are derived from n = 2 biological replicates.
(TIF)

**S8 Fig. Cell viability of ALKBH1 modified cells.** A and B. HEK293T cells were transfected with siRNAs targeting ALKBH1 (Negative control "ctrl": scrambled siRNA) or ALKBH1-pcDNA3 (Negative control "ctrl": empty vector) for 48 h. The cell viability was then measured by MTT assay based on the OD value. The mean cell viability was calculated. The cell viability in ctrl. was considered 100% to compare the cell viability in ALKBH1 siRNA or ALKBH1-pcDNA3 transfected cells. C. Cell viability was assessed using the MTT assay, based on OD values for the cycloheximide chase experiment where HEK293T cells were transfected with siRNAs targeting ALKBH1 (Negative control "ctrl": scrambled siRNA).
(TIF)

**S9 Fig. Glucose-dependent demethylation activity of ALKBH1. A-C.** Longer incubation times (24 h) corresponding to Figure 4. Cells were processed and analyzed by (A) RT-qPCR, and total cell lysates were analyzed by (B; 8 h and C; 24 h) western blotting to assess ALKBH1 expression under varying glucose and serum conditions. D. In parallel, total tRNA was enzymatically digested and subjected to LC-MS/MS for the detection of RNA modifications. Error bars represent the mean ± SEM of three or six biological replicates. Statistical significance was determined using one-way ANOVA without correction. (Created in BioRender. Henzeler, B. (2025).).
(TIF)

**S10 Fig. Uncropped gel images of SDS-PAGE analysis of purified ALKBH1, METTL8 and EcMiaA shown in Supporting Fig S1.** A-D. Human ALKBH1 and ALKBH1I218A, METTL8 and E. coli MiaA were overexpressed and purified from E. coli. E. Human ALKBH1 was overexpressed and purified from HEK293T cells.
(TIF)

**S11 Fig. Uncropped gel images of Urea-PAGE analysis of in vitro transcribed tRNA.** A) shown in Fig 2A and B) Supporting Fig S3.
(TIF)

**S12 Fig. Uncropped blot images of western blots shown in Supporting Fig S5B, D and E (A-E) and Supporting Fig S9B and C (F and G).**
(TIF)

**S1 Table. Protein sequences.** Highlighted in bold and underlined are additional amino acid residues. Ile218 mutated to prevent auto-hydroxylation (ALKBH1I218A) is highlighted in red. His 231 and Asp 233 in the active site, coordinating the iron were mutated to inactivate ALKBH (ALKBH1AxA), are highlighted in blue.
(PDF)

**S2 Table. Used Nucleoside digestion mix.**
(PDF)

**S3 Table. tRNA sequences. Highlighted in bold is the C in the anticodon arm that is methylated by METTL8.**
(PDF)

**S4 Table. qPCR primers.**
(PDF)

**S5 Table. siRNA sequences.**
(PDF)

**S6 Table. Biotinylated capture probes for tRNA molecules.**
(PDF)

**S7 Table. Antibodies used in this study.**
(PDF)

**S8 Table. Abbreviations of the specified RNA modifications are listed below.**
(PDF)

## Author contributions

**Conceptualization:** Bennett Henzeler, Ken Kögel, Lena J. Daumann, Thomas Carell, Stefanie Kaiser, Sabine Schneider.

**Data curation:** Bennett Henzeler, Sabine Schneider.

**Formal analysis:** Bennett Henzeler, Olga Hofmeister, Ken Kögel, Yuyang Qi, Felix Hagelskamp, Matthias Heiß.

**Funding acquisition:** Thomas Carell, Stefanie Kaiser, Sabine Schneider.

**Investigation:** Bennett Henzeler, Olga Hofmeister, Ken Kögel, Yuyang Qi, Felix Hagelskamp, Matthias Heiß, Florian Schelter, Felix Xu, Sabine Schneider.

**Methodology:** Bennett Henzeler, Olga Hofmeister, Ken Kögel, Yuyang Qi, Felix Hagelskamp, Matthias Heiß, Florian Schelter, Felix Xu.

**Project administration:** Stefanie Kaiser, Sabine Schneider.

**Resources:** Thomas Carell.

**Supervision:** Lena J. Daumann, Stefanie Kaiser.

**Validation:** Lena J. Daumann.

**Visualization:** Bennett Henzeler.

**Writing – original draft:** Stefanie Kaiser.

**Writing – review & editing:** Lena J. Daumann, Stefanie Kaiser, Sabine Schneider.

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
