## [Decision Letter · Decision Letter 0]

6 Jan 2026

Dear Dr. Schneider,

plosone@plos.org. . . . A letter that responds to each point raised by the academic editor and reviewer(s). You should upload this letter as a separate file labeled 'Response to Reviewers'.A marked-up copy of your manuscript that highlights changes made to the original version. You should upload this as a separate file labeled 'Revised Manuscript with Track Changes'.An unmarked version of your revised paper without tracked changes. You should upload this as a separate file labeled 'Manuscript'.

We look forward to receiving your revised manuscript.

Kind regards,

Mohammad H. Ghazimoradi

Academic Editor

PLOS One

Journal Requirements:

https://journals.plos.org/plosone/s/file?id=ba62/PLOSOne_formatting_sample_title_authors_affiliations.pdf....

“Deutsche Forschungsgemeinschaft (SCHN 1273-9) and SFB 1309 (project number 325871075) BMFTR Cluster4Future program (Cluster for Nucleic Acid Therapeutics Munich, CNATM, ID: 03ZU1201AA).”

“This work was financially supported by the Deutsche Forschungsgemeinschaft (SCHN 1273-9) and SFB 1309 (project number 325871075) to S.S., L.D., T.C. and S.K., as well as the BMBF Cluster4Future program (Cluster for Nucleic Acid Therapeutics Munich, CNATM, ID: 03ZU1201AA). “

“Deutsche Forschungsgemeinschaft (SCHN 1273-9) and SFB 1309 (project number 325871075) BMFTR Cluster4Future program (Cluster for Nucleic Acid Therapeutics Munich, CNATM, ID: 03ZU1201AA).”

Reviewers' comments:

Reviewer's Responses to Questions

**Comments to the Author**

1. Is the manuscript technically sound, and do the data support the conclusions?

Reviewer #1: Yes

Reviewer #2: Partly

2. Has the statistical analysis been performed appropriately and rigorously?

Reviewer #1: Yes

Reviewer #2: Yes

3. Have the authors made all data underlying the findings in their manuscript fully available?

Reviewer #1: Yes

Reviewer #2: Yes

4. Is the manuscript presented in an intelligible fashion and written in standard English?

Reviewer #1: Yes

Reviewer #2: Yes

Reviewer #1: Henzeler et al present their analysis of the potential and controversial ALKBH1 function as an RNA modifying enzymes. This subject is timely and important, and their negative results are a very important addition to the literature. ALKBH1 has been a subject of controversy for quite some time. Initially reported to protect against DNA alkylation damage (Drabløs et al, DNA Repair, 2004), ALKBH1 was later recognized as important RNA modifying enzyme. First, Liu et al identified it as an m1A demethylase (Liu et al, Cell, 2016). This was later validated by 2 independent groups. Rashad et al (RNA Biol, 2020) showed that ALKBH1 demethylates m1A specifically during oxidative stress. Arguello et al (Nat Comm, 2022) also examined the m1A activity using orthogonal methods. Surprisingly, Xie et al, (Cell, 2018) reported that ALKBH1 is a DNA m6A demethylase, a claim that was thoroughly challenged in multiple articles (Douvlataniotis et al, Sci Adv, 2020; Lyu et al, Cell Discov, 2022). ALKBH1 is also thoroughly validated as a tRNA dioxygenase, catalyzing the oxygenation of wobble m5C to hm5C and then to f5C (Kawarada et al, Nucleic Acids Res, 2017, Arguello et al, Nat Comm, 2022; Nakayashiki et al, Biorxiv, 2025; Shen et al, Cancer Discov, 2025). Through this effect, ALKBH1 can regulate the decoding of rare leucine codons and drive oncogenesis (Nakayashiki et al and Shen et al). Nonetheless, due to its clear importance in cancer biology, many claims have been made regarding its function, using suboptimal methods. For example, researchers claimed that it demethylates m7G, m5C, and m1A in mRNA, to mention a few. This ambiguity is indeed detrimental to understanding its function and role in diseases, which is why the article at hand is important and timely, especially with the great interest in RNA modifications currently observed.

There are, nonetheless, several points that need to be taken into consideration. Given that the authors report negative results, engaging with the different approaches reported in the literature regarding ALKBH1 activity is important. Additionally, there are a couple of new important articles on ALKBH1 function that engage with these inquiries that need to be cited, as they support the authors’ observations. Below I will give specific comments:

1-In the abstract, the authors write: “In addition, varying the glucose and fetal bovine serum (FBS) concentration in the growth medium of HEK293T cells, in combination with alkbh1 siRNA-mediated knock-down, also shows no impact on the tRNA modification spectra”. This is a bit of generalization, given that ALKBH1 does in fact impact wobble modifications (f5C and hm5C) that are not characterized here. It is better to change the language to either focus on methylation modifications or ALKBH1 eraser activity.

2-The authors’ in vivo analysis focused on specific mitochondrial tRNAs that were isolated using complementary DNA oligos (mt-tRNA-SerUCA and mt-tRNA-ThrUGU). However, why the authors did not attempt to evaluate other reported mt-tRNAs that interact with ALKBH1 such as mt-tRNA-Arg and mt-tRNA-Lys (reported in Kawarada et al, Nucleic Acids Res, 2017). Additionally, the article by Liu et al (Cell, 2016) focused heavily on cytosolic tRNA-Met, which the authors did not test. Validating previously reported ALKBH1-interacting tRNAs is of utmost importance.

3-The authors did not observe changes in tRNA methylation status after exposing HEK293T cells to different glucose or FBS concentrations. However, the authors only cultured the cells for 8 hours, which might not be enough to elicit responses at the level of tRNA modifications. Furthermore, simple changes in glucose or FBS concentrations might not be sufficient, unless protracted durations, to induce an effect. Rashad et al (RNA Biol, 2020) reported that ALKBH1 acts as m1A demethylase during acute oxidative stress. Thus, a more stressful stimulus, such as FBS or glucose deprivation, could elicit the response. In a preprint (Rashad et al, 2024, doi: https://doi.org/10.1101/2024.02.14.580389) it was shown that FBS deprivation does impact m1A levels. Thus, I would recommend either extending the duration of cell culture (24~48 hours) or using deprivation media to make sure that there is no effect (ideally various conditions should be used).

4-Another point related to the Rashad et al, RNA Biol, 2020 report. Testing under conditions of oxidative stress could also be of value, given that Arguello et al (Nat Comm, 2022) validated stress induced demethylase activity of ALKBH1.

5- There are 2 new articles that I would like to direct the authors’ attention to: Shen et al, Cancer Discov, 2025, in which the authors show that ALKBH1 is a tRNA dioxygenase acting to facilitate the translation of rare Leucine codons. The second paper is Nakayashiki et al, Biorxiv, 2025 (doi: https://doi.org/10.1101/2025.11.28.691056). This preprint is important also as it not only validates the function of ALKBH1 on wobble modifications and rare leucine codons, but also the authors perform CRISPR/Cas9 KO or ALKBH1 and overexpression in 5 different cancer cell lines and use LC-MS/MS to analyze the changes in tRNA modifications. The authors did not observe demethylation activity of ALKBH1 in their analysis, which further validates the presented work.

6-Final note, I recommend the authors to engage more with the wealth of aberrantly reported activities of RNA modifying enzymes. Discussing the importance of rigorous testing and the differences between in vitro and in vivo observations should be further highlighted in the discussion. Many articles have reported “new RNA modifying functions” of various enzymes using suboptimal tools such as simple dot blots. These trends should be challenged and addressed. It is important to also discuss the previous reports on ALKBH1 tRNA activity by highlighting the methods used and their strengths/weaknesses.

Reviewer #2: Summary and overall assessment

In this study, Bennett Henzeler et al. aim to reconcile ALKBH1’s reported breadth of in vitro RNA demethylation activity with the frequent difficulty of detecting in vivo effects under standard culture conditions. Using quantitative isotope-dilution nucleoside LC–MS/MS the authors demonstrate in vitro reduction of multiple modified nucleosides (m1A, m3C, m5C) in total tRNA extracts and develop an enzymatic pipeline to generate defined mt-tRNA substrates carrying m3C32 (and i6A37) and observe partial m3C reduction upon ALKBH1 treatment. They also report no detectable changes in bulk nucleoside modification profiles upon ALKBH1 overexpression or siRNA knockdown in HEK293T cell nor after 8 h glucose/FBS variation in HEK293T/HeLa cells for two cytosolic tRNAs.

The in vitro biochemical work is competent, and the negative cellular results are potentially valuable. However, multiple conclusions are currently over-scoped relative to the experimental design, and key “dependency” claims (Fe(II)/α-KG) are not directly demonstrated. Most importantly, the authors’ own data indicate ALKBH1 has an unusually long half-life, which substantially weakens inference from short-term perturbations (siRNA and 8 h metabolic shifts). The study can have some improvements before it can be published. See the comments below:

Major comments

1) Cellular “no effect” conclusions are underpowered given ALKBH1 stability; siRNA alone is not a strong depletion strategy here. The authors show ALKBH1 abundance “does not change over 72 h after cycloheximide” and interpret this as an unusually long half-life. They also acknowledge that remaining ALKBH1 activity may be sufficient to maintain steady-state modification levels. Under these circumstances, “no detectable change” after siRNA is not strong evidence that ALKBH1 lacks cellular substrates. Complement siRNA with a protein degradation approach (e.g., PROTAC-mediated ALKBH1 degradation) in both HEK293T and HeLa cells to achieve rapid, deep depletion at the protein level, and re-test the key modification readouts. At minimum, the authors should quantify ALKBH1 protein reduction (not just mRNA) under each perturbation regime and correlate depletion depth with modification changes (or lack thereof).

2) After the overexpression of ALKBH1, the author should perform the localization assay of the ALKBH1, since it can both locate in the mitochondria, nucleus and cytosolic ER.

Minor comment:

1) “m3C32 is a major natural substrate” is over-stated relative to the cellular evidence. The manuscript concludes that the enzymatically generated substrates show ALKBH1 reduces m3C by ~25–40% and therefore “m3C32 is indeed a major natural substrate”, yet endogenous mt-tRNA modification spectra remain unchanged upon overexpression/knockdown in HEK293T cells. It is better to tone down wording to “biochemically competent substrate in vitro,” unless the authors can demonstrate a cellular condition where ALKBH1 depletion/activation produces a measurable, reproducible change.

2) Fe (II)/α-KG dependency is asserted but not directly tested. The manuscript states an intent to “further evaluate the Fe (II)/α-KG dependency of ALKBH1 in vitro”, but the described Succinate-Glo experiments primarily validate that succinate generation correlates with catalytic competency and is partially confounded by auto-hydroxylation, rather than establishing cofactor (Fe (II) /α-KG) dependency. The author should include explicit –Fe(II) and –α-KG conditions (and ideally chelation, e.g., EDTA or titration of Fe concentration;) for this assay. Alternatively, the authors may change their claim and tune down it. The currently assay is more like a validation of the success of the catalytic death mutant of ALKBH1.

3) The 8 h glucose/FBS assay does not robustly test “glucose does not impact ALKBH1-mediated RNA modification”. The authors culture cells for 8 h under different glucose/FBS conditions and observe no change in m1A in two enriched cytosolic tRNAs, plus only minor effects on alkbh1 mRNA. Given the long ALKBH1 half-life and the likelihood that bulk tRNA modification pools turnover slowly, an 8 h window is not a stringent test of glucose-driven, ALKBH1-dependent remodeling. I suggest reframing the claim narrowly (no detectable change under this 8 h condition in these two cytosolic tRNAs)

4) Please specify clearly which ALKBH1 overexpression construct was used in each cellular experiment (untagged vs C-terminal Strep-tag)

5) In the method section, the tRNA refolding part (line 383) please cite a reference. In addition, if the author can add evidence to show the tRNA refolding correctly, that would be appreciated.

.

Reviewer #1: **Yes:** Sherif RashadSherif RashadSherif RashadSherif Rashad

Reviewer #2: No

---

## [Author Response · Author response to Decision Letter 1]

2 Feb 2026

Review Comments to the Author

We thank the reviewer for thoroughly reading of our manuscript and the positive and constructive comments, that allowed us to markedly improve the manuscript.

Reviewer #1: Henzeler et al present their analysis of the potential and controversial ALKBH1 function as an RNA modifying enzymes. This subject is timely and important, and their negative results are a very important addition to the literature. ALKBH1 has been a subject of controversy for quite some time. Initially reported to protect against DNA alkylation damage (Drabløs et al, DNA Repair, 2004), ALKBH1 was later recognized as important RNA modifying enzyme. First, Liu et al identified it as an m1A demethylase (Liu et al, Cell, 2016). This was later validated by 2 independent groups. Rashad et al (RNA Biol, 2020) showed that ALKBH1 demethylates m1A specifically during oxidative stress. Arguello et al (Nat Comm, 2022) also examined the m1A activity using orthogonal methods. Surprisingly, Xie et al, (Cell, 2018) reported that ALKBH1 is a DNA m6A demethylase, a claim that was thoroughly challenged in multiple articles (Douvlataniotis et al, Sci Adv, 2020; Lyu et al, Cell Discov, 2022). ALKBH1 is also thoroughly validated as a tRNA dioxygenase, catalyzing the oxygenation of wobble m5C to hm5C and then to f5C (Kawarada et al, Nucleic Acids Res, 2017, Arguello et al, Nat Comm, 2022; Nakayashiki et al, Biorxiv, 2025; Shen et al, Cancer Discov, 2025). Through this effect, ALKBH1 can regulate the decoding of rare leucine codons and drive oncogenesis (Nakayashiki et al and Shen et al). Nonetheless, due to its clear importance in cancer biology, many claims have been made regarding its function, using suboptimal methods. For example, researchers claimed that it demethylates m7G, m5C, and m1A in mRNA, to mention a few. This ambiguity is indeed detrimental to understanding its function and role in diseases, which is why the article at hand is important and timely, especially with the great interest in RNA modifications currently observed.

There are, nonetheless, several points that need to be taken into consideration. Given that the authors report negative results, engaging with the different approaches reported in the literature regarding ALKBH1 activity is important. Additionally, there are a couple of new important articles on ALKBH1 function that engage with these inquiries that need to be cited, as they support the authors’ observations. Below I will give specific comments:

We have included in the introduction additional references to emphasis further on the ambiguity of ALKBH1 function.

1-In the abstract, the authors write: “In addition, varying the glucose and fetal bovine serum (FBS) concentration in the growth medium of HEK293T cells, in combination with alkbh1 siRNA-mediated knock-down, also shows no impact on the tRNA modification spectra”. This is a bit of generalization, given that ALKBH1 does in fact impact wobble modifications (f5C and hm5C) that are not characterized here. It is better to change the language to either focus on methylation modifications or ALKBH1 eraser activity.

We have reworded the sentence, which reads now: “In addition, varying the glucose and fetal bovine serum (FBS) concentration in the growth medium of HEK293T cells, in combination with alkbh1 siRNA-mediated knock-down, also shows no impact on tRNA methylation.”

2-The authors’ in vivo analysis focused on specific mitochondrial tRNAs that were isolated using complementary DNA oligos (mt-tRNA-SerUCA and mt-tRNA-ThrUGU). However, why the authors did not attempt to evaluate other reported mt-tRNAs that interact with ALKBH1 such as mt-tRNA-Arg and mt-tRNA-Lys (reported in Kawarada et al, Nucleic Acids Res, 2017). Additionally, the article by Liu et al (Cell, 2016) focused heavily on cytosolic tRNA-Met, which the authors did not test. Validating previously reported ALKBH1-interacting tRNAs is of utmost importance.

Our selection of mitochondrial tRNAs was guided by 3 considerations, which are now stated in the manuscript: (i) we find activity of ALKBH1 in vitro towards m3C. As in vivo activity of ALKBH1 against m3C-containing mt-tRNAs were not yet reported, mt-tRNA-Ser and mt-tRNA-Thr were our main candidates of interest. (ii) Both mt-tRNA-Ser and mt-tRNA-Thr contain m1A and may be substrate to ALKBH1 as reported by Liu et al for cytosolic tRNAs (Cell 2016) and (iii) m5C/f5C and f5Cm-modified mt-tRNAs (from Kawarada et al, NAR, 2017) were not considered as these are not technologically accessible with our LC-MS method. Therefore mt-tRNA-Met was not studied. Sequence and modification of mt-tRNASer (UCN and AGY) and mt-tRNAThr as reported by Suzuki et al, 2020, Nat Commun.

Regarding the suggested mt-tRNAs of the reviewer, we want to point out that mt-tRNAs are of extreme low abundance and thus their isolation requires either specialized equipment as shown by the Suzuki lab in the past years (Kawarada, NAR, 2017 or the hallmark study Suzuki et al, 2020, Nat Commun) or a highly sensitive LC-MS set-up that allows analysis of sub-nanogram amounts of mt-tRNA isolates (this manuscript). Nevertheless, the abundance of mt-tRNAs remains a challenge and therefore, 3 mt-tRNA isoacceptors was the maximum of tRNAs we could isolate with the material gained after economic and technical considerations. Our priority was guided by the following considerations: The reported mt-tRNA-Met against which ALKBH1 shows activity would have been the ideal control. Yet, the substrate/product of ALKBH1, namely m5C/f5C, in mt-tRNA-Met, cannot be analyzed in our ammonium acetate chromatography set-up, and thus this potential control could not be done. Further, to the best of our knowledge Liu et al (Cell 2016) analyzed cytosolic tRNAs Lys and Arg and not mt-tRNAs. In conjunction with our observation that ALKBH1 targets m3C in vitro, our highest priority was the isolation of the 2 m3C-containing tRNAs and the remaining seryl-tRNA with a C32 shown in the manuscript. These also contain m1A which expands the study of Liu et al (Cell 2016), although we did not observe in vivo activity.

3-The authors did not observe changes in tRNA methylation status after exposing HEK293T cells to different glucose or FBS concentrations. However, the authors only cultured the cells for 8 hours, which might not be enough to elicit responses at the level of tRNA modifications. Furthermore, simple changes in glucose or FBS concentrations might not be sufficient, unless protracted durations, to induce an effect. Rashad et al (RNA Biol, 2020) reported that ALKBH1 acts as m1A demethylase during acute oxidative stress. Thus, a more stressful stimulus, such as FBS or glucose deprivation, could elicit the response. In a preprint (Rashad et al, 2024, doi: https://doi.org/10.1101/2024.02.14.580389) it was shown that FBS deprivation does impact m1A levels. Thus, I would recommend either extending the duration of cell culture (24~48 hours) or using deprivation media to make sure that there is no effect (ideally various conditions should be used).

In order to address this, we incubated HEK293T cells under reduced FBS and/or glucose conditions for 24h. However it should be noted that after 24h under these stress condition cell confluency was lower compared to standard growth conditions, albeit we did not observe increased cell death (MTT assay). It should be noted, however, that HEK293T cells are a highly transformed, immortalized cell line, and most other cells would be unlikely to tolerate such stress conditions over this time frame. Under these circumstances, for example during the onset of cell death, numerous factors beyond ALKBH1 may become up- or downregulated or activated or inactivated, which could also potentially influence epigenetic modifications. However, we did not observe any significant impact on ALKBH1 and nucleoside modifications in total tRNA.

We agree with the reviewer, that to identify the conditions and cellular states in which the clearly important role of ALKBH1 plays surfaces. However, this is beyond the scope of our study, where we emphasis that highly immortalized cells under standard laboratory culturing conditions are not the appropriate model system to achieve this.

4-Another point related to the Rashad et al, RNA Biol, 2020 report. Testing under conditions of oxidative stress could also be of value, given that Arguello et al (Nat Comm, 2022) validated stress induced demethylase activity of ALKBH1.

We agree with the reviewer that identifying conditions, cell states etc where ALKBH1 activity is important is of paramount interest to elucidate the exact role of this clearly important enzyme. However, this is beyond the scope of this manuscript. Here we show that the cell culture conditions and cell system chosen is essential and that ALKBH1 does not play a role in the demethylation of tRNAs in human cells lines grown under optimal conditions.

5- There are 2 new articles that I would like to direct the authors’ attention to: Shen et al, Cancer Discov, 2025, in which the authors show that ALKBH1 is a tRNA dioxygenase acting to facilitate the translation of rare Leucine codons. The second paper is Nakayashiki et al, Biorxiv, 2025 (doi: https://doi.org/10.1101/2025.11.28.691056). This preprint is important also as it not only validates the function of ALKBH1 on wobble modifications and rare leucine codons, but also the authors perform CRISPR/Cas9 KO or ALKBH1 and overexpression in 5 different cancer cell lines and use LC-MS/MS to analyze the changes in tRNA modifications. The authors did not observe demethylation activity of ALKBH1 in their analysis, which further validates the presented work.

We thank the reviewer for pointing this out and have included these reports in the discussion.

6-Final note, I recommend the authors to engage more with the wealth of aberrantly reported activities of RNA modifying enzymes. Discussing the importance of rigorous testing and the differences between in vitro and in vivo observations should be further highlighted in the discussion. Many articles have reported “new RNA modifying functions” of various enzymes using suboptimal tools such as simple dot blots. These trends should be challenged and addressed. It is important to also discuss the previous reports on ALKBH1 tRNA activity by highlighting the methods used and their strengths/weaknesses.

We thank the reviewer for this excellent suggestion and we have re-written the discussion to first discuss our observations on ALKBH1 activity observed by us and others. As the final paragraph, we discuss the challenges in the in vivo analysis of RNA demethylation detection and highlight suitable technologies to approach proving in vivo activity of demethylases.

Reviewer #2: Summary and overall assessment

In this study, Bennett Henzeler et al. aim to reconcile ALKBH1’s reported breadth of in vitro RNA demethylation activity with the frequent difficulty of detecting in vivo effects under standard culture conditions. Using quantitative isotope-dilution nucleoside LC–MS/MS the authors demonstrate in vitro reduction of multiple modified nucleosides (m1A, m3C, m5C) in total tRNA extracts and develop an enzymatic pipeline to generate defined mt-tRNA substrates carrying m3C32 (and i6A37) and observe partial m3C reduction upon ALKBH1 treatment. They also report no detectable changes in bulk nucleoside modification profiles upon ALKBH1 overexpression or siRNA knockdown in HEK293T cell nor after 8 h glucose/FBS variation in HEK293T/HeLa cells for two cytosolic tRNAs.

The in vitro biochemical work is competent, and the negative cellular results are potentially valuable. However, multiple conclusions are currently over-scoped relative to the experimental design, and key “dependency” claims (Fe(II)/α-KG) are not directly demonstrated. Most importantly, the authors’ own data indicate ALKBH1 has an unusually long half-life, which substantially weakens inference from short-term perturbations (siRNA and 8 h metabolic shifts). The study can have some improvements before it can be published. See the comments below:

For AlkB-family of dioxygenases their Fe(II)/α-KG dependency has been clearly established. The residues involved in binding to Fe(II) and α-KG are well conserved in the AlkB homologs and are essential for catalytic activity. We introduced two point mutations of the crucial Asp residues, which abolishes activity as seen in the succinate glow assay.

We have now carried out additional experiments and tested incubating HEK293T cells under FBS and/or glucose starvation conditions for 24h. Please see below detailed comment below and above (Reviewer 1).

Major comments

1) Cellular “no effect” conclusions are underpowered given ALKBH1 stability; siRNA alone is not a strong depletion strategy here. The authors show ALKBH1 abundance “does not change over 72 h after cycloheximide” and interpret this as an unusually long half-life. They also acknowledge that remaining ALKBH1 activity may be sufficient to maintain steady-state modification levels. Under these circumstances, “no detectable change” after siRNA is not strong evidence that ALKBH1 lacks cellular substrates. Complement siRNA with a protein degradation approach (e.g., PROTAC-mediated ALKBH1 degradation) in both HEK293T and HeLa cells to achieve rapid, deep depletion at the protein level, and re-test the key modification readouts. At minimum, the authors should quantify ALKBH1 protein reduction (not just mRNA) under each perturbation regime and correlate depletion depth with modification changes (or lack thereof).

We have also analysed the effect on siRNA-mediated knockdown of ALKBH1 on the protein level by Westerblot (Supporting Fig. S5). Here we see a significant reduction on protein level. We have now also included Westernblots for cells grown under low FBS and/or low glucose (new Supporting Fig S9).

A recent manuscript by Nakayashiki et al (bioRxiv preprint doi: 10.1101/2025.11.28.691056) where ALKBH1 was knocked-out using CRISPR-Cas9 also showed no effect on f5C-levels in tRNA in cell lines. This further supports the key massage of our manuscript that in human cells lines grown under optimal conditions ALKBH1 does not play an important role in the demethylation of tRNAs.

Of course it is of paramount interest to identify the specific cellular states and conditions where ALKBH1 activity is of importance. However, this is beyond the scope of the present manuscript.

2) After the overexpression of ALKBH1, the author should perform the localization assay of the ALKBH1, since it can both locate in the mitochondria, nucleus and cytosolic ER.

In a previous study it was shown that upon overexpression ALKBH1 localises to the mitochondria in several human cell lines such as HeLa, HT-29 and PC3 cells (Wagner et al 2019). We have added the following sentence to the manuscript: “It was previously shown that upon overexpression ALKBH1 locates to the mitochondria in several human cell lines [19].”

Minor comment:

1) “m3C32 is a major natural substrate” is over-stated relative to the cellular evidence. The manuscript concludes that the enzymatically generated substrates show ALKBH1 reduces m3C by ~25–40% and therefore “m3C32 is indeed a major natural substrate”, yet endogenous mt-tRNA modification spectra remain unchanged upon overexpression/knockdown in HEK293T cells. It is better to tone down wording to “biochemically competent substrate in vitro,” unless the authors can demonstrate a cellular condition where ALKBH1 depletion/activation produces a measurable, reproducible change.

We re-worded the sentence: “These results show that ALKBH1 reduces m3C methylation levels in these tRNAs by about 25-40 % (Fig 2B) and that m3C32 is indeed a substrate of ALKBH1.”

2) Fe (II)/α-KG dependency is asserted but not directly tested. The manuscript states an intent to “further evaluate the Fe (II)/α-KG dependency of ALKBH1 in vitro”, but the described Succinate-Glo experiments primarily validate that succinate generation correlates with catalytic competency and is partially confounded by auto-hydroxylation, rather than establishing cofactor (Fe (II) /α-KG) dependency. The author should include explicit –Fe(II) and –α-KG conditions (and ideally chelation, e.g., EDTA or titration of Fe conc

---

## [Decision Letter · Decision Letter 1]

8 Feb 2026

Dear Dr. Schneider,

Thank you for submitting your manuscript to PLOS ONE. After careful consideration, we feel that it has merit but does not fully meet PLOS ONE’s publication criteria as it currently stands. Therefore, we invite you to submit a revised version of the manuscript that addresses the points raised during the review process.

We look forward to receiving your revised manuscript.

Kind regards,

Yu-Hsuan Tsai

Academic Editor

PLOS One

**Journal Requirements:**

**Additional Editor Comments:**

Somehow the previous handling editor is no longer available, and I have taken over the processing of your revised submission (PONE-D-25-59513_R1). Both reviewers appear generally satisfied with the revisions. However, please **still address Reviewer 1’s second point** before we proceed further.

In addition, please **carefully proofread the revised manuscript for grammar, typographical errors, and internal consistency**. Several issues are listed below as examples; please correct these and ensure there are no similar errors elsewhere. I anticipate that these changes should be straightforward and should not require substantial time.

As I will be on vacation **12–23 February** for the Chinese New Year holiday, if you can return the revised files **by 11 February**, I should be able to complete processing before I am away. No further external review is expected for the next round.

### Examples of required language/consistency corrections

**Abstract / Results**

* “does not impact on rRNA modifications” → revise to “does not impact rRNA modifications” or “has no impact on rRNA modifications”.

* “siRNA-mediated knock-down of alkbh1” → standardize to “siRNA-mediated knockdown of ALKBH1” (and ensure consistent capitalization throughout).

* “ALKBH1 locates to the mitochondria” → revise to “ALKBH1 localizes to mitochondria” (add a comma after the introductory clause, as appropriate).

* “may be substrate to ALKBH1” → revise to “may be substrates for ALKBH1”.

* “LC-MS set-up.The …” → add missing space: “set-up. The …”.

**Methods: siRNA transfection**

* “siRNA mediated” → “siRNA-mediated”.

* “following manufacturer protocol” → “following the manufacturer’s protocol”.

* “… vortexing. jetPRIME® reagent …” → revise to “... vortexing. **The** jetPRIME® reagent …”.

* “1-s vortex” → revise to “vortexed for 1 s” (or “1 s vortexing”).

**tRNA isoacceptor purification**

* “reverse complementary” → use “reverse-complementary” or “reverse complement”.

* “DNA oligonucleotide were” → “DNA oligonucleotides were”; also add article: “from **a** previous publication”.

* “Equal amount” → “Equal amounts” (and adjust verb agreement where needed).

* “15 μl” → “15 μL”.

* “Each iso-acceptor was purified as triplicate.” → “Each isoacceptor was purified in triplicate.” (also standardize “isoacceptor” vs. “iso-acceptor” throughout).

**Immunoblotting**

* “over night” → “overnight”.

* “Equal amount of protein were …” → “Equal amounts of protein were …” (or “An equal amount … was …”).

* “sulfate” vs “sulphate” in the SDS-PAGE description → standardize to one spelling (preferably “sulfate”).

* “following manufacturers protocol” → “following the manufacturer’s protocol”.

**Miscellaneous**

* “stored at -80°C until further analysis After defrosting,” → revise to “…analysis. After defrosting, …”.

* Author contributions: “und” → “and”.

* Cycloheximide chase: “mock were considered 100%” → “mock was considered 100%” (or “mock samples were considered 100%”).

Reviewers' comments:

Reviewer's Responses to Questions

**Comments to the Author**

Reviewer #1: All comments have been addressed

Reviewer #2: All comments have been addressed

2. Is the manuscript technically sound, and do the data support the conclusions?

Reviewer #1: Yes

Reviewer #2: Yes

3. Has the statistical analysis been performed appropriately and rigorously?

Reviewer #1: Yes

Reviewer #2: Yes

4. Have the authors made all data underlying the findings in their manuscript fully available?

Reviewer #1: Yes

Reviewer #2: Yes

5. Is the manuscript presented in an intelligible fashion and written in standard English?

Reviewer #1: Yes

Reviewer #2: Yes

Reviewer #1: The authors responded to my comments in an excellent manner. I agree with their assessment of ALKBH1 function, especially in light of new evidence. I have 2 minor comments:

1- IN Nakayashiki et al's preprint, they used 4 cell lines and not 5 (This was a typo in my previous comment, but the authors should double check)

2- The authors should add a limitation section in their discussion including some of the points they discussed (mitochondrial tRNA limitations, technology limitations in assessing other modifications, different results compared to literature due to different approaches, etc.)

Apart from those 2 minor points, I believe my concerns and comments are all addressed.

Reviewer #2: In this revision, the authors strengthen their manuscript by (1) adding protein-level validation of ALKBH1 knockdown/overexpression and documenting an unusually long ALKBH1 half-life, (2) extending glucose/FBS stress experiments to 24 h with matched ALKBH1 mRNA/protein and tRNA nucleoside LC–MS/MS readouts, (3) adding a –α-KG control and rewording prior “Fe(II)/α-KG dependency” language, and (4) providing DLS evidence supporting in vitro tRNA refolding. In summary, these additional experiments has addressed my previous concerns and improved the rigor of this study. They also appropriately narrow several over-scoped claims. The paper is suitable for publication.

.

Reviewer #1: No

Reviewer #2: No

---

## [Author Response · Author response to Decision Letter 2]

9 Feb 2026

We thank again the reviewer for their time and thoughtful comments that have allowed us to further improve our manuscript.

Comments to the Author

1. If the authors have adequately addressed your comments raised in a previous round of review and you feel that this manuscript is now acceptable for publication, you may indicate that here to bypass the “Comments to the Author” section, enter your conflict of interest statement in the “Confidential to Editor” section, and submit your "Accept" recommendation.

Reviewer #1: All comments have been addressed

Reviewer #2: All comments have been addressed

2. Is the manuscript technically sound, and do the data support the conclusions?

Reviewer #1: Yes

Reviewer #2: Yes

3. Has the statistical analysis been performed appropriately and rigorously?

Reviewer #1: Yes

Reviewer #2: Yes

4. Have the authors made all data underlying the findings in their manuscript fully available?

Reviewer #1: Yes

Reviewer #2: Yes

5. Is the manuscript presented in an intelligible fashion and written in standard English?

Reviewer #1: Yes

Reviewer #2: Yes

6. Review Comments to the Author

Reviewer #1: The authors responded to my comments in an excellent manner. I agree with their assessment of ALKBH1 function, especially in light of new evidence. I have 2 minor comments:

1- IN Nakayashiki et al's preprint, they used 4 cell lines and not 5 (This was a typo in my previous comment, but the authors should double check)

In the cited manuscript by Nakayashiki et al 5 cell lines were used. However, 4 are only relevant for our manuscript. Thus we have corrected it in the manuscript.

2- The authors should add a limitation section in their discussion including some of the points they discussed (mitochondrial tRNA limitations, technology limitations in assessing other modifications, different results compared to literature due to different approaches, etc.)

We have added the following limitation section to the discussion: “In this context quantitative LC-MS analysis must be considered the gold standard in nucleoside detection due to its high specificity and sensitivity, that other detection methods such as dot blots cannot provide. …. Nevertheless, a main challenge for detecting and quantifying modified ribonucleosides is the reproducibility across laboratories [40]. Here variations in growth conditions, cell lines, RNA isolation and processing as well as lack of a standard form of modified RNA for researchers to use as a benchmark for validating their studies hamper the accurate quantification of RNA modifications. In addition, there is no single optimal method for all 170 known modified ribonucleosides or the more than 50 human RNA modification.”

Apart from those 2 minor points, I believe my concerns and comments are all addressed.

We thank the reviewer for his time and valuable comments.

Reviewer #2: In this revision, the authors strengthen their manuscript by (1) adding protein-level validation of ALKBH1 knockdown/overexpression and documenting an unusually long ALKBH1 half-life, (2) extending glucose/FBS stress experiments to 24 h with matched ALKBH1 mRNA/protein and tRNA nucleoside LC–MS/MS readouts, (3) adding a –α-KG control and rewording prior “Fe(II)/α-KG dependency” language, and (4) providing DLS evidence supporting in vitro tRNA refolding. In summary, these additional experiments has addressed my previous concerns and improved the rigor of this study. They also appropriately narrow several over-scoped claims. The paper is suitable for publication.

We thank the reviewer for his time and valuable comments.

---

## [Editor Report · Decision Letter 2]

13 Feb 2026

ALKBH1 activity in vitro and human cell lines by isotope dilution mass spectrometry

PONE-D-25-59513R2

Dear Dr. Schneider,

We’re pleased to inform you that your manuscript has been judged scientifically suitable for publication and will be formally accepted for publication once it meets all outstanding technical requirements.

Kind regards,

Yu-Hsuan Tsai

Academic Editor

PLOS One
---

## [Editor Report · Acceptance letter]

PONE-D-25-59513R2

PLOS One

Dear Dr. Schneider,

I'm pleased to inform you that your manuscript has been deemed suitable for publication in PLOS One. Congratulations! Your manuscript is now being handed over to our production team.

Kind regards,

on behalf of

Dr. Yu-Hsuan Tsai

Academic Editor

PLOS One